# Diel streamflow cycles suggest more sensitive snowmelt-driven streamflow to climate change than land surface modeling does

Sebastian A. Krogh[1,2,3], Lucia Scaff[4], Gary Sterle[2], James W. Kirchner[5,6], Beatrice Gordon[1], and Adrian Harpold[1,2]

[1]Department of Natural Resources and Environmental Science, University of Nevada, Reno, 89557, USA

[2]Global Water Center, University of Nevada, Reno, 89557, USA

[3]Departamento de Recursos Hídricos, Facultad de Ingeniería Agrícola, Universidad de Concepción, Chillán, 3812120, Chile

[4]Global Water Futures, Canada First Research Excellence Fund (CFREF), University of Saskatchewan, Saskatoon, SK S7N 3H5, Canada.

[5]Department of Environmental Systems Science, ETH Zurich, CH-8092 Zurich, Switzerland

[6]Swiss Federal Research Institute WSL, CH-8903 Birmensdorf, Switzerland

*Correspondence to*: Sebastian A. Krogh (skrogh@udec.cl)

**Abstract.** Climate warming will cause mountain snowpacks to melt earlier, reducing summer streamflow and threatening water supplies and ecosystems. Quantifying how sensitive streamflow timing is to climate change, and where it is most sensitive, remains a key question. Physically based hydrological models are often used for this purpose; however, they have embedded assumptions that translate into uncertain hydrological projections that need to be quantified and constrained to provide reliable inferences. The purpose of this study is to evaluate differences in projected changes to streamflow volume timing by the end of the century between a new empirical model based on diel (daily) streamflow cycles and regional land-surface simulations across the mountainous western US. We develop an observational technique for detecting streamflow responses to snowmelt using incoming solar radiation and diel cycles of streamflow to detect when snowmelt occurs. We measure the date of the $20^{th}$ percentile of snowmelt days ($DOS_{20}$), across 31 watersheds affected by snow in the western US, as a proxy for the beginning of snowmelt-initiated streamflow. Historic $DOS_{20}$ varies from mid-January to late May, with warmer sites having earlier snowmelt-mediated streamflow. Mean annual $DOS_{20}$ strongly correlates with the dates of 25% and 50% annual streamflow volume ($DOQ_{25}$ and $DOQ_{50}$, both $R^2 = 0.85$), suggesting that a one-day earlier $DOS_{20}$ corresponds with a one-day earlier $DOQ_{25}$ and 0.7-day earlier $DOQ_{50}$. Empirical projections of future $DOS_{20}$ based on a multiple linear regression across sites and years under the RCP8.5 scenario for the late $21^{st}$ century show that $DOS_{20}$ will occur on average $11\pm4$ days earlier per 1°C of warming; however, $DOS_{20}$ in colder watersheds (mean November-February air temperature, $T_{NDJF}$ < -8°C) is on average 70% more sensitive to climate change than in warmer watersheds ($T_{NDJF}$ > 0°C). Moreover, empirical projections of $DOQ_{25}$ and $DOQ_{50}$ based on projected $DOS_{20}$ changes are about four and two times more sensitive, respectively,

to earlier streamflow than those simulated by a state-of-the-art land surface model (NoahMP-WRF) under the same scenario. Given the importance of changing streamflow timing for water resources, and the significant discrepancies found in projected streamflow sensitivity, snowmelt detection methods such as $DOS_{20}$ based on diel streamflow cycles may help to constrain parameter selection and improve hydrological predictions.

## 1    Introduction

The role of earlier snowmelt in driving earlier streamflow timing is of great concern in a changing climate (Barnett et al., 2005; Harpold and Brooks, 2018; Musselman et al., 2017; Stewart et al., 2004, 2005). Earlier winter and spring streamflow volume comes at the expense of later summer streamflow in regions like the western US (Hidalgo et al., 2009; McCabe and Clark, 2005; Regonda et al., 2005; Stewart et al., 2004, 2005) and challenges reservoir operations (Barnett et al., 2005; Immerzeel et al., 2020; Viviroli et al., 2011). Furthermore, ecosystems may evaporate more water as reductions in albedo increase energy inputs (Meira Neto et al., 2020), decreasing runoff from upland forested watersheds (Foster et al., 2016; Jepsen et al., 2018; Milly and Dunne, 2020). More than 50% of mountainous watersheds play essential roles in supporting downstream systems (Viviroli et al., 2007) and snowpack changes are likely to increase lowland agriculture water stress (Immerzeel et al., 2020). However, it remains difficult to predict how much streamflow timing and amount will shift in future climates due to altered snow accumulation patterns (Mote et al., 2018), melt rates (Musselman et al., 2017), and shifts from snowfall to rainfall (Klos et al., 2014).

Due to the complexity of upland streamflow generation, physically based hydrological models are typically used to predict how snowpack changes will interact with the critical zone (CZ), and thus affect short-term flood behavior and seasonal water supply forecasts (Kopp et al., 2018; Wood and Lettenmaier, 2006). In mountainous regions like the western United States (US), models need to accurately simulate snow processes across watersheds with varying snowpack conditions (Serreze et al., 1999) and then transport and store that water in the CZ along hillslopes and watersheds with varying subsurface properties (Brooks et al., 2015). More precipitation falling as rain instead of snow will result in streamflow dynamics that more closely mirror the timing of rainfall. Precipitation phase is mediated by basin elevation and hypsometry (Jennings et al., 2018; Wayand et al., 2015), which also influences precipitation amounts (Houze, 2012), with higher elevations and steeper watersheds typically having higher precipitation and snowfall. Solar radiation is the primary energy source for snowmelt in snow-dominated montane watersheds (Cline, 1997; Marks and Dozier, 1992), explaining the importance of cloudiness in regulating snowmelt and streamflow processes, as evidenced by negative correlations between cloud cover and melt rates (Sumargo and Cayan, 2018). Shallower snowpacks have less cold content and begin their melt earlier when solar radiation is lower (Harpold et al., 2012; Harpold and Brooks, 2018; Musselman et al., 2017), which shifts streamflow earlier (Clow, 2010). Storage and drainage of water in the CZ control the sensitivity of streamflow to earlier rain or melt water inputs. For example, snowmelt-mediated spring streamflow timing is more sensitive to climate change in watersheds with rapid subsurface drainage than in landscapes with deep groundwater reservoirs that drain slowly (Safeeq et al., 2013). In contrast, the sensitivity of snowmelt-

mediated summer streamflow volume to climate change has shown to be higher in slow-draining watersheds (Tague and Grant, 2009). The complexity of these storage relationships is exemplified by isotopic evidence showing that the fraction of

streamflow that is "young water" (less than three months old) is smaller in steeper watersheds (Jasechko et al., 2016), suggesting that interactions between CZ water storage and changing hydrometeorology will be challenging to predict in mountainous areas. In a recent data-driven review, Gordon et al. (2022) proposed a predictive framework composed of three testable and inter-related mechanisms to infer changes to snowmelt-driven streamflow response under warming. Such mechanisms are associated with snow season energy and mass exchanges, the intensity of snow season liquid water input and

the synchrony of energy and water availability, and their analysis highlights the complexities in predicting changes to streamflow in regions where multiple mechanisms interact.

Hydrologists typically apply two types of modeling tools to predict future streamflow: empirical models and more mechanistically oriented models (conceptual or physically based land surface models). Empirical models assume that long-term and often site-to-site statistical relationships among predicting variables (e.g., precipitation and air temperature) and water

fluxes (e.g., evapotranspiration and streamflow) can be used to understand and model their likely changes over time or space. Empirical models used to predict changes over time (sometimes referred to space-for-time substitutions) have been used in fields such as hydrology (Goulden and Bales, 2014; Jepsen et al., 2018; Sivapalan et al., 2011), biodiversity (Blois et al., 2013) and tree growth (Klesse et al., 2020) to predict responses to climate change. Such models use information from different places ("space"), typically spanning a wide range of conditions (e.g., climate gradient), to predict changes over time. For example,

observed characteristics from warm regions maybe used to infer future changes in cold regions due to global warming. A limitation of this approach is that it neglects non-correlated (or independent) changes in spatially varying factors (Jepsen et al., 2018). For example, heterogeneous patterns of warming, variations in precipitation and vegetation, or changes that occur at different temporal scales (e.g., soil properties versus rain-snow line transition) are implicitly neglected in such empirical frameworks. Conversely, physically based models embed physics and state-of-the-art understanding of hydrological processes.

These models typically require some degree of calibration or validation to observations (e.g., daily streamflow) to increase and assess their predictive skill. The current generation of regional weather models using the Weather Research and Forecasting model (WRF) (Skamarock et al., 2008) coupled to the Noah Multi Parameterization land surface model (Noah-MP) (Niu et al., 2011) has shown promising results for modeling atmospheric and snow processes in the contiguous US (He et al., 2019; Liu et al., 2017; Musselman et al., 2017; Scaff et al., 2020). For example, snow simulations have been used to quantify

mountain snowmelt and streamflow response to climate change (Musselman et al., 2017, 2018). These simulations use a pseudo global warming approach, which perturbs the historical climate with a climate change signal from an ensemble of global climate models (GCMs); using this perturbation avoids systemic biases in the GCMs and avoids issues related to their interannual variability (Liu et al., 2017). Given the importance of snowmelt to streamflow generation and its uncertain sensitivity to climate change, new tools that allow for comparisons between land surface models and empirically based

predictions of future streamflow are valuable and could help to diagnose modeling issues that imped better predictions.

Few simple, low-cost observational tools exist to separate rainfall-driven from snowmelt-driven contributions to streamflow or to separate this year's melt from the previous years' melt and storage. One method that can be straightforwardly applied to existing long-term observations is based on coupled diel cycles in solar radiation, snowmelt, and streamflow (Kirchner et al., 2020; Lundquist and Cayan, 2002). Diel (24-hours) cycles in streamflow and shallow groundwater levels are often found in mountainous systems driven by snow and ice melt, and evapotranspiration, which are both ultimately driven by solar radiation inputs (Kirchner et al., 2020). This mechanistic response has been used to study watershed properties like kinematic wave celerity (Kirchner et al., 2020), the impact of snowpack variability on streamflow timing (Lundquist and Dettinger, 2005), groundwater fluctuations (Loheide and Lundquist, 2009), and transitions from snowmelt to evapotranspiration-dominated streamflow fluctuations (Kirchner et al., 2020; Mutzner et al., 2015; Woelber et al., 2018). More recently, Kirchner et al. (2020) combined local observations and remote sensing to show that streamflow diel response was tightly controlled by the timing of snowpack disappearance. However, it remains unknown whether information embedded in the diel streamflow response following snowmelt events can be used to inform streamflow predictions due to climate change, and whether such projections are consistent with current land-surface simulations. The purpose of this research is to evaluate potential differences in projected changes to streamflow volume timing by the end of the century between a new empirical diel streamflow-based model and regional land-surface simulations across mountainous western US headwater catchments. To this aim, we extend the 'diel cycle index' approach of Kirchner et al. (2020) using diel streamflow observations to detect the occurrence of days when streamflow is coupled to snowmelt inputs, and investigate their contributions to historical variability in streamflow volume timing. We then compare empirical diel streamflow-based projections by the end of the century under an RCP8.5 pseudo global warming scenario against predictions from a state-of-the-art land surface model (under the same climate scenario) across 31 mountainous watersheds in the western US to answer the following questions:

1. Does the historical diel streamflow-based analysis show earlier snowmelt in warmer watersheds and years, and can we use the observed timing of snowmelt to predict the timing of streamflow volume?

2. Where is the timing of snowmelt the most sensitive to climate change as projected by an empirical diel streamflow-based model?

3. Do historical streamflow volume timings and future empirical diel streamflow-based projections diverge from commonly used, state-of-the-art land surface models?

A list with the abbreviations used in this study is presented in Table 1.

## 2    Methods

### 2.1    Study Domain and Data

We studied snowmelt-driven streamflow in 31 mountainous watersheds in the western US (Table 2), spanning snow fractions of 0.27 to 0.78 (Figure A3A), aridity index values from 0.22 to 2.86 (Addor et al., 2017), and soil depths from 0.27 to 2.52 m (Addor et al., 2017; Pelletier et al., 2016) (Table 2). These watersheds are part of the CAMELS (Catchments Attributes and MEteorology for Large-sample Studies) dataset (Addor et al., 2017; Newman et al., 2015), which provides daily streamflow

and meteorological forcing, among other observed and simulated hydrometeorological variables at the watershed scale. These

130 watersheds were chosen because their streamflows are unregulated, they have relatively small drainage areas ($< 250$ km$^2$), and they are at relatively high elevations ($> 1,000$ masl). This last criterion was introduced to focus on watersheds with snowmelt-driven streamflow regimes. The names, locations, elevations, slopes, drainage areas and other key characteristics of the 31 watersheds are presented in Table 2.

The data used in this analysis include hourly streamflow and incoming shortwave radiation, and mean daily relative humidity, air temperature and precipitation. Hourly streamflow was obtained from the US Geological Survey. Hourly incoming shortwave radiation is from phase 2 of the National Land Data Assimilation System (NLDAS-2) (Xia et al., 2012) at the nearest grid point to the watershed outlet. Mean daily relative humidity, air temperature and precipitation at the watershed scale are from CAMELS, based on the DAYMET dataset (daymet.ornl.gov), which in turn is interpolated from existing ground

observations. Available hourly streamflow records vary significantly across watersheds, extending back to 1986 for some sites. Figure A1A shows the number of years that have more than 70, 80 and 90% of days with hourly records for the period between December 1 and August 1. Based on this preliminary analysis, we selected water years with more than 80% of days with hourly streamflow records. This threshold for data availability results in most watersheds having more than 5 years to analyze (except for sites #10 and #30 with 4 years).

**2.2  Snowmelt and Streamflow Diel Coupling**

To infer the occurrence of days when solar radiation-driven snowmelt is coupled to the streamflow, hereafter referred as snowmelt days for simplicity, we calculated the correlation between hourly values of solar radiation and lagged streamflow (Figure 1). A snowmelt day is defined as a day in which the Spearman correlation between hourly solar radiation and lagged streamflow is statistically significant (p-value$\leq$0.01) and exceeds a given cutoff. Due to the lagged diel streamflow response

after snowmelt, we lagged diel streamflow from solar radiation between 6 and 18 hours, computed the correlation of all combinations, and kept those statistically significant correlations that were above a pre-defined correlation cutoff. Although having both a correlation cutoff and a statistical significance criterion may be redundant, we used both to guarantee significant correlations above different correlation cutoffs. We tried several correlation cutoffs (r>0.5, 0.6, 0.7, 0.8 and 0.9; see Figure 1 for r>0.6) to assess their effects on the detection algorithm (Figure A2). The preliminary lag window of 6 to 18 hours was used

to avoid confounding snowmelt signals with evapotranspiration (ET)-induced streamflow diel responses (Kirchner et al., 2020; Mutzner et al., 2015; Woelber et al., 2018). ET-induced streamflow diel response can positively correlate with solar radiation with lags below 6 hours due to the previous day's ET, and above 18 hours due to the next day's ET diurnal signal (Kirchner et al., 2020). However, this preliminary lag window may incorrectly select days with a rainfall-induced streamflow diel response or rain-on-snow events. To minimize this, we further restricted the lags that could be selected based on optimum lags

from snowmelt days with clear skies. Clear-sky days were defined as days with solar radiation greater than 80% of the clear-sky solar radiation (grey areas in left panels on Figure 1). This lag window was defined on a monthly and watershed basis and

was calculated as the lags between the 10th and 90th percentile of clear-sky days with Spearman correlations above 0.8. This second filter also helped to avoid the incorrect selection of ET-induced streamflow diel response, as it minimized the chance of selecting 18-hr lags that can be associated with ET. Despite efforts to select only snowmelt-driven streamflow diel responses, this methodology does not guarantee that rainfall-driven streamflow diel changes with lags within our lag window will always be excluded. Excluding such cases would require hourly precipitation observations, which are not available for all of our study watersheds. However, we believe that any such cases will minimally affect the results of our analysis.

To provide a better idea of the potential impact that rainfall may have on our proposed diel analysis, particularly on the effect of rain-on-snow events, we analyzed which days classified as snowmelt days also had rainfall. We assessed daily rainfall using the daily precipitation time series from CAMELS based on the DAYMET product for each watershed. A false detection rate metric was computed for each watershed, in which every day classified as a snowmelt with daily precipitation above 5 mm and a mean daily air temperature above 2 °C was assumed to be mis-classified (Figure 2). A false detection rate of 100% means that all snowmelt days were mis-classified and 0% means that no days had significant rainfall. On average, the false detection rate was estimated at 7% with a standard deviation of 5%, and only watersheds #24 and #31 (located in WA and OR, respectively) are above 15%, with 21% and 29%, respectively. This suggest that the effect of potential rainfall-induced diel streamflow cycle (including rain-on-snow events) in most watersheds is low (except for watersheds #24 and #31), supporting further analysis. We also assessed the mean cross-site false detection rate for precipitation thresholds of 1 mm and 10 mm and found reasonable values of 12% and 3%, respectively. However, we believe that 1 mm is not a reasonable threshold as a 1 mm rainfall event is unlikely to produce a distinguishable diel streamflow signal or could represent error/noise in the DAYMET product.

## 2.3    The empirical diel streamflow-based model

We defined the day when the 20th percentile of the snowmelt days occurs ($DOS_{20}$) as a new metric to characterize the seasonality of early snowmelt for each water year and watershed. However, other metrics such as the 5th, 10th and 30th percentiles (presented in the appendices) were also investigated to assess the impact of this choice on the analysis. We chose this metric because we expected it to be associated with the timing of streamflow volume, and that the choice of slightly earlier or later snowmelt days would not substantially change our results. We fitted a stepwise multiple linear regression model (MLR, p-value<0.01, Equation 1) to reconstruct historical $DOS_{20}$ across all our sites (Figure A4) using four climate variables as predictors: total precipitation, air temperature, relative humidity, and solar radiation.

$$DOS_{20} = \beta_1 x_1 + \beta_2 x_2 + \beta_3 x_3 + \beta_4 x_4 + \beta_5 x_1 x_2 + \beta_6 x_1 x_3 \qquad (1)$$
$$+ \beta_7 x_1 x_4 + \beta_8 x_2 x_3 + \beta_9 x_2 x_4 + \beta_{10} x_3 x_4$$

Where $x_1$ is cumulative air temperature (°C), $x_2$ is cumulative precipitation (mm), $x_3$ is mean relative humidity (%), $x_4$ is mean solar radiation (W m$^{-2}$), and $\beta_i$ are the regression coefficients. Mean annual climate variables were calculated for the period

between November $1^{st}$ and $DOS_{20}$. This results in $DOS_{20}$ being present in both sides of Equation 1; therefore, the stepwise MLR requires an iterative solution when used in a predictive mode (i.e., for the climate change analysis when $DOS_{20}$ is unknown). The MLR model is the basis of our empirical diel streamflow-based model, which is used to assess changes in

$DOS_{20}$ due to climate change (i.e., changes in $x_1$, $x_2$, $x_3$ and $x_4$ in Eq. (1)). We verified the stepwise MLR assumptions, namely, linear relationships between each predictor and $DOS_{20}$, normally distributed residuals, homoscedasticity, and absence of strong multicollinearity (as suggested by a Variance Inflation Factor $< 3$). We also tested other metrics related to the timing of early snowmelt events. These included: the first snowmelt day, the first three consecutive snowmelt events, and the $5^{th}$, $10^{th}$ and $30^{th}$ percentiles of snowmelt days ($DOS_5$, $DOS_{10}$ and $DOS_{30}$, respectively). All metrics were also computed using each of the

different Spearman correlation cutoffs (Table A1, A2, A3, A4 and A5), but the main analysis presented here focuses on $DOS_{20}$ based on snowmelt days calculated with hourly Spearman correlations $>0.8$.

We predict changes to $DOS_{20}$ based on the stepwise MLR model and an end-of-the-century mean climate change signal from WRF (Liu et al., 2017). WRF was run under a high emission scenario (RCP8.5) using the pseudo global warming approach for the end of the century. Overall, it projects a warmer ($4 – 5.2°C$), wetter (0 - 20% increase in precipitation) climate (Figure

A4 and A5). These mean annual changes in climate were applied to the predictors in the stepwise MLR model to predict changes in $DOS_{20}$. As previously mentioned, predictors used in the stepwise MLR were calculated for the period between November $1^{st}$ and $DOS_{20}$; therefore, as we do not know the value of $DOS_{20}$ in the future, an iterative solution is required to solve for $DOS_{20}$ in Equation 1. We find a numerical solution using a 2-day convergence threshold between iterations, so that $|DOS20_{i+1} – DOS20_i| \leq 2$ days, where '$i$' is the number of the iteration.

## 2.4    Streamflow Volume Timing from a Land-Surface Model

Historical NoahMP-WRF simulations include the period 2001-2013 over the contiguous US at 4-km spatial resolution, and the period 2071-2100 under pseudo global warming (Liu et al., 2017). NoahMP-WRF simulations include an improved Noah configuration aiming to better represent the snow physics. These improvements include (Liu et al., 2017): the rain-snow transition is based on a microphysics partitioning approach as opposed to a subjective temperature-based approach, patchy

snowpack are allowed in the calculation of the surface energy balance, the heat transport from rainfall to the ground is included, and the snow depletion curve is vegetation-dependent. These improvements allow for a better representation of the surface energy balance, and the simulation of snow accumulation and melt processes. We used daily watershed-scale outputs of surface and subsurface runoff from historical and future NoahMP-WRF simulations to estimate $DOQ_{25}$ and $DOQ_{50}$. Given the range of the watershed drainage areas (4 - 236 $km^2$, Table 2), watersheds covering several grid cells use the total surface and

subsurface runoff for their corresponding grid cells. Small watersheds are represented by only the single nearest NoahMP-WRF grid cell. The way NoahMP-WRF is implemented within WRF lacks a streamflow routing scheme such as the one in WRF-Hydro (Gochis et al., 2020); therefore, we used the sum of surface and subsurface runoff to estimate $DOQ_{25}$ and $DOQ_{50}$. We also repeated the analysis using surface runoff only, leading to similar results (Figure A7). Given the relatively coarse

NoahMP-WRF spatial resolution (4 km) compared to the watershed drainage areas (4 - 236 km$^2$), we assume that mean streamflow timing metrics are not significantly affected by the lack of streamflow routing.

## 3    Results

### 3.1    Empirical Relationships Between DOS$_{20}$, Climate and Streamflow

Mean DOS$_{20}$ has a strong regional variability that is reasonably captured by a negative linear correlation ($R^2 = 0.48$) with the mean winter air temperature (November to February, T$_{NDJF}$) in watersheds with T$_{NDJF}$<-3°C, whereas warmer watersheds do not follow the same pattern (Figure 3A and Figure 4A). Warmer sites (T$_{NDJF} > $-3 °C) have a more variable mean DOS$_{20}$ ranging from mid-January to early May, whereas the coldest sites (T$_{NDJF}$ <-8°C) have a later and less variable DOS$_{20}$ around mid to late May. On average, the regression suggests that a 1 °C of warming results in 7.2-day earlier DOS$_{20}$. The relationship between later DOS$_{20}$ and colder T$_{NDJF}$ is also found in the year-to-year variations in DOS$_{20}$ at most watersheds (21 out of the 31), with warmer years experiencing earlier DOS$_{20}$ (Figure 3B). A strong linear relationship was found between the date of the 25% of the annual streamflow volume (DOQ$_{25}$) and T$_{NDJF}$. Warmer watersheds (T$_{NDJF}$>0°C) generate streamflow the earliest (between mid-December and early March) compared to the coldest watersheds (T$_{NDJF}$<-8°C), with DOQ$_{25}$ between early and late May (Figure 3C). On average, the cross-site regression shows that a 1°C increase in T$_{NDJF}$ produces a 13-day earlier DOQ$_{25}$. For most watersheds (25 out of 31), interannual regressions show a similar pattern with warmer years having earlier DOQ$_{25}$; however, these interannual regressions have shallower slopes than the cross-site relationship (Figure 3B and 3D). Previous work by Stewart et al. (2005) also related seasonal meteorological patterns with the spring onset and streamflow timing, and found similar relationships (e.g., warmer watersheds have earlier spring onset and streamflow timing). However, the definition of the spring onset was based on the cumulative hydrograph (the day when the cumulative departure from the mean streamflow was the minimum), as opposed to our more mechanistic diel streamflow analysis. Other definitions for spring onset based on streamflow, snow pillows and air temperature are presented by Lundquist et al. (2004).

Strong correlations between DOS$_{20}$ and both DOQ$_{25}$ and DOQ$_{50}$ (the dates at which 25% and 50% of the annual streamflow volume are reached) ($R^2 = 0.85$, Figure 5A and 5C) suggest connections between the timing of snowmelt and streamflow generation across watersheds and years. On average, sites that melt earlier are associated with earlier DOQ$_{25}$ (Figure 5A) and a lower ratio of snowfall to total precipitation (snow fraction<0.5). The relationship between DOS$_{20}$ and DOQ$_{25}$ closely follows the 1:1 line (Figure 5A), although three sites in Washington and Oregon (sites #24, #25 and #31, see Table 2 and Figure 6A) deviate substantially from this pattern, perhaps because they receive relatively little of their precipitation as snow. Similar watershed-level relationships using interannual variability in DOQ$_{25}$ were found for most watersheds, with statistically significant slopes varying between 0.4 and 2.5 day day$^{-1}$ (Figure 5B). DOS$_{20}$ also predicts DOQ$_{50}$ well, with 10-day earlier snowmelt producing 7-day earlier DOQ$_{50}$ on average (Figure 5C), and similar watershed-level interannual relationships (Figure

5D). The same three relatively rainy watersheds have $DOQ_{50}$ prior to the $DOS_{20}$ (Figure 5C and Figure 6B), suggesting that early snowmelt timing is not an important predictor of $DOQ_{50}$ in such places.

## 3.2     Diel Streamflow-Based Sensitivity of Snowmelt Timing ($DOS_{20}$) to Climate Change

We fitted a stepwise MLR with four climate variables (air temperature, precipitation, relative humidity, and solar radiation) to predict the diel streamflow-based $DOS_{20}$ metric across watersheds and years. A total of 333 watershed-year combinations of

$DOS_{20}$ and climate variables were used to train the stepwise MLR model. The watershed-year relationship between observed and MLR predictions has a relatively high $R^2$ of 0.83, a root mean square error (RMSE) of 17.5 days, and normally distributed residuals ($p < 0.01$) off the 1:1 line and centered at 0 with a standard deviation of 17.3 days (Figure 7A). The relationship between observations and MLR predictions of inter-watershed mean annual $DOS_{20}$ (Figure 7B) is also strong ($R^2 = 0.83$ and RMSE = 13.2 days) and follows the 1:1 line. Similarly, when we look at interannual values, represented by the lines

overlapping the circles in Figure 7B, we find a good agreement with most slopes close to 1:1 (see inset plot Figure 7B). This analysis demonstrates that the MLR model can reasonably represent both the mean annual $DOS_{20}$ values at each watershed and their interannual variability. Table A4 shows standardized beta coefficients that indicate the importance of each climate variable in the stepwise MLR. For the 0.8 correlation cutoff we found that incoming shortwave radiation has the greatest importance (beta = 0.75), followed by relative humidity (beta = 0.37) and air temperature (beta = -0.31).

Empirical diel streamflow-based projections under climate change show earlier mean annual $DOS_{20}$ in all watersheds, with significant variability from site to site (Figure 8A). Most watersheds show significant end-of-century changes in $DOS_{20}$ ranging from up to three months earlier in cold sites where, historically, snowmelt under clear-sky conditions dominates (circles in Figure 8A), to as little as 20 days earlier in warm sites under historically cloudier conditions. The cross-site average change in

$DOS_{20}$ is 55.3 days with a standard deviation of 21.8 days. In many watersheds the mean projection of $DOS_{20}$ under climate change is within the historically observed variability in $DOS_{20}$ (Figure 8A). The empirical model predicts that colder watersheds ($T_{NDJF} \leq -8°C$) on average are about 70% more sensitive to climate change ($13.7 \pm 4.6$ days $°C^{-1}$) than warmer watersheds are ($T_{NDJF} > 0°C$) ($8.1 \pm 6.2$ day $°C^{-1}$), as represented by the change in the $DOS_{20}$ per degree of warming (Figure 8B). Site #24 (South Fork Tolt River, WA.) shows almost no change in its $DOS_{20}$, which can be attributed to its weaker climate change

signal compared to the other watersheds (about +4°C, 5% precipitation increase, and virtually no change in humidity and solar radiation; Figure A4). When we look at the mean sensitivity across all watersheds, the diel streamflow-based analysis suggest an average sensitivity of $11.1 \pm 4.2$ days $°C^{-1}$.

### 3.3 Sensitivity of Streamflow Timing to Climate Change: Empirical diel streamflow-based model versus NoahMP-WRF

We compared historical and empirical diel streamflow-based projections for $DOQ_{25}$ and $DOQ_{50}$ with those from NoahMP-WRF. Empirical streamflow timing sensitivity projections for $DOS_{20}$ under climate change were built using the linear regressions presented in Figure 5A and 5C ($DOQ_{25}$ and $DOQ_{50}$ vs $DOS_{20}$) with projected changes in $DOS_{20}$ using the MLR under climate change. Empirical projections for $DOQ_{25}$ range from early January to late May (red symbols, Figure 9A), advancing between 20 and 100 days under RCP 8.5 (x-axis, Figure 9C). The $DOQ_{50}$ is projected to advance between roughly

15 and 65 days (x-axis, Figure 9D), ranging from mid-February to late May (red symbols, Figure 9B). The historical $DOQ_{25}$ is greatly underestimated by NoahMP-WRF (blue symbols, Figure 9A) with a mean $DOQ_{25}$ in mid-February, whereas historical $DOQ_{25}$ is in early April (50-day mean difference). Projected changes to $DOQ_{25}$ by NoahMP-WRF under pseudo global warming range between early January to mid-March (mean in early February), whereas empirical diel streamflow-based projections range between early January and late March (mean in mid-February; Figure 9A). These results indicate that

empirical diel streamflow-based projections of $DOQ_{25}$ are about four times more sensitive to climate change than those from NoahMP-WRF ($\Delta DOQ_{25}$ averages about -60 days for empirical model and -15 days for NoahMP-WRF; Figure 9C). Historical $DOQ_{50}$ is reasonably well represented by NoahMP-WRF under the current climate (blue symbols, Figure 9B) with a mean difference of only 7 days, but future changes of about -20 days are roughly half of the -40 days predicted by the empirically based projections (Figure 9D). Empirical diel streamflow-based projections of $DOQ_{50}$ range between mid-February and early

April, whereas NoahMP-WRF projections range between mid-March and mid-May. Watersheds with the largest disagreement between the empirical model and NoahMP-WRF projections for streamflow volume timing are those where $DOS_{20}$ is the most sensitive to climate change, represented by the orange and yellow symbols in Figure 9C and 9D. These watersheds are characterized by historical cold winter temperatures ($T_{NDJF}<-6°C$) with snowmelt occurring mostly under sunny conditions (circle symbols) and are mostly located in the Rocky Mountains.

## 4    Discussion

The new $DOS_{20}$ metric represents the timing of early snowmelt-mediated streamflow based on the diel streamflow fluctuations and suggests that shifts in snowmelt timing in colder, sunnier watersheds have a greater effect on streamflow volume timing than in warmer, cloudier watersheds where snowmelt is more interspersed with rain. Despite the intuitive connections between

snowmelt and streamflow, empirically linking changes in earlier snowmelt rates (Harpold and Brooks, 2018; Musselman et al., 2017) with changes in streamflow amount (Barnhart et al., 2016) and timing (Stewart et al., 2004) has been challenging (Weiler et al., 2018). This is in part due to the scales at which snow (point-scale) and streamflow (watershed-scale) are typically measured. For example, evidence of snowmelt at Snow Telemetry (SNOTEL) locations in the US has shown that snowmelt events are more intermittent at sites with higher humidity, and future modeling suggests slower, earlier snowmelt in the largest

snowpacks in areas with lower humidity and cloud cover (Harpold and Brooks, 2018; Musselman et al., 2017). However, the potential cascading effects of earlier and slower snowmelt on streamflow amount and timing are relatively unexplored (e.g. Berghuijs et al., 2014). Not surprisingly, the warmest and cloudiest watersheds have lower snow fractions and a more rainfall-dominated streamflow regime, and thus have less (and often no) interannual correlation between $DOS_{20}$ and the metrics $DOQ_{25}$ and $DOQ_{50}$ (Figure 5A and 5C), illustrating the limitations of the diel streamflow method in rain-dominated watersheds; as

also suggested by the false detection rate analysis (Figure 2) in watersheds #24 and #31 in Washington and Oregon, respectively. Rain-on-snow events are particularly challenging to detect with our analysis, as days with a lower percentage of incoming shortwave radiation (<80% of clear-sky) are filtered out to avoid issues with potential rainfall-dominated diel signals. Conversely, the colder and sunnier watersheds, primarily in the intermountain region, have strong interannual correlations between $DOS_{20}$ and $DOQ_{25}$ (Figure 5A and Figure 6A), reflecting the importance of snowmelt (instead of rain) in controlling

streamflow volume timing. We currently lack physically based representations of many processes linking snowpack storage, snowmelt, subsurface storage, and the timing of water release following a hydrologic event (i.e., snowmelt or rainfall event). Snowmelt modeling in complex terrain is challenged by steep climate gradients and by the lack of adequate forcing data required to run models. Characterizing precipitation phase and timing in steep watersheds remains challenging in rain-to-snow transition zones (Harpold et al., 2017; Jennings et al., 2018; Wayand et al., 2015), which will presumably increase in extent in

the future (Klos et al., 2014). Complex terrain has a large effect on radiation fluxes, which are hard to capture at kilometer spatial scales (Müller and Scherer, 2005) used in some land surface models. Nonetheless, this issue is less important in warmer, cloudier watersheds where longwave radiation and sensible heat are larger components of the energy balance (Mazurkiewicz et al., 2008). Forests exert a strong control on the snowpack mass and energy balance (Lundquist et al., 2013; Pomeroy et al., 1998; Safa et al., 2021) with spatially heterogeneous effects on snow accumulation and melt that remain challenging to model

(Broxton et al., 2015; Krogh et al., 2020). The presence of preferential flowpaths through the snowpack impacts the timing of melt release (Leroux and Pomeroy, 2017) and is not typically included in hydrological models. Once snowmelt is released from the snowpack, simulating (and validating) what fraction flows as subsurface and surface runoff remains difficult. Decades of tracer studies (e.g., Godsey et al., 2010; Kirchner, 2003) have shown that streamflow during and after hydrologic events

(i.e., snowmelt or rainfall events) is typically 'old water' that has been stored in the watershed for months to years. Land surface models like NoahMP-WRF lack realistic groundwater stores to represent old water and are at spatial resolutions that make hillslope and near-stream processes difficult to represent (Fan et al., 2019). For example, previous work at Sagehen Creek (site #23) suggests that streamflow remains ~80% groundwater even during the snowmelt freshet (Urióstegui et al., 2017). Innovative observations and/or analyses that give new physical insights, like the diel streamflow analysis, can be used to derive such hydrologic representations, which could improve our prediction of hydrological systems (Kirchner, 2006).

Because the diel streamflow analysis does not require the many assumptions that are embedded in physically based models, it is an independent tool that can be used to verify historical streamflow simulations from sub-daily resolved hydrological models. For example, land surface models could be benchmarked against observed snowmelt days based on the diel streamflow analysis or metrics like the $DOS_{20}$, aiming to better represent processes associated with snowmelt-driven streamflow generation. The diel streamflow analysis is also easier to implement than detailed process-based catchment models because it only requires observed hourly streamflow data and solar radiation. Solar radiation can be reliably represented by land surface models like NLDAS-2 (Luo et al., 2003) that assimilate field observations and remotely sensed radiation (including the effects of clouds) into an atmospheric modeling framework. We tested the sensitivity of some modeling decisions, such as the correlation cutoff between hourly solar radiation and streamflow used to detect snowmelt days and metrics for snowmelt timing, and found similar sensitivities of $DOS_{20}$ to climate change across different correlation cutoffs and snowmelt timing percentiles (Table A5). Metrics like the first snowmelt day or the first three consecutive snowmelt days showed less consistent results (Table A5), likely due to individual early or mid-winter melt events that do not necessarily represent the seasonal watershed behavior. The diel streamflow analysis has four main limitations that need to be examined in future work: (1) it requires a steep enough stage-discharge relationship that daily streamflow cycles can be detected across the flow regime, (2) it focuses on snowmelt driven by solar radiation (and energy fluxes synchronized with it), (3) it is sensitive to assumptions about the lag time between solar radiation and streamflow, and (4) it is sensitive to assumptions about evapotranspiration losses. A steep stage-discharge relationship, in which small changes in discharge are associated with large changes in stage, is ideal to observe small diel streamflow changes with sufficient precision. Another assumption is that the majority of snowmelt is correlated with solar radiation. This assumption is supported by the importance of solar radiation in process-based studies of maritime and continental snowpacks (Cline, 1997; Jepsen et al., 2012; Marks and Dozier, 1992). Because our method allows the lag time between solar radiation and streamflow to vary within a predefined window, we expect it to capture other important energy fluxes like sensible heat that often lag the diel patterns of solar radiation by several hours (Ohmura, 2001). This approach is not suitable for capturing rain-on-snow events, which are most common in maritime watersheds, but also occur in continental settings (Musselman et al., 2018). It may also misclassify rainfall-driven diel streamflow cycles, although we checked for rainfall-induced cycles and found that these are, on average, a small fraction (7%, Figure 2). In rainier watersheds (lower snow fraction), our analysis may be more uncertain than in watersheds with a more snowfall-dominated regime. Nonetheless, the relationships between streamflow timing (i.e., $DOS_{20}$, $DOQ_{25}$ and $DOQ_{50}$) and meteorological drivers in rainier sites showed

cross-site and interannual relationships that are consistent with those in colder, more snow-dominated places (except for watersheds #24, #25 and #31) (e.g., Figure 3A and 3C). The third limitation is that the spatiotemporal variability in snowpack, surface and subsurface storage, and evapotranspiration will change the magnitude and lag time of the diel streamflow response (Kirchner et al., 2020; Lundquist and Cayan, 2002; Lundquist and Dettinger, 2005), which we address by allowing variable watershed- and month-specific time lags. However, lag times greater than 24 hours, which are associated with large watersheds or large subsurface storage, will make this method impossible to apply. Also, the method may miss early snowmelt-driven diel cycles in watersheds with dry soils, as the diel signal will be buffered by the subsurface storage capacity before generating a measurable streamflow response. The fourth limitation is that evapotranspiration losses must be small relative to snowmelt inputs, which is necessary because the effect of evapotranspiration is out of phase with the effect of snowmelt (Kirchner et al., 2020). Evapotranspiration effects are minimized by focusing on early snowmelt when evapotranspiration losses are often assumed to be small (Bowling et al., 2018; Cooper et al., 2020; Winchell et al., 2016).

Previous empirically based implementations have been used to predict catchment-scale sensitivity of snowmelt-driven streamflow to changing climate using observations (Berghuijs et al., 2014; Stewart et al., 2005) and historical model outputs (Barnhart et al., 2016). The empirical diel streamflow model based on the stepwise MLR suggests that humidity explains roughly as much or more variation in $DOS_{20}$ than temperature does (Table A4), and that solar radiation explains about twice as much $DOS_{20}$ variation as either humidity or temperature does. This is consistent with an energy budget dominated by solar radiation (Marks and Dozier, 1992), but also with a coupling between humidity and latent heat and longwave radiation effects (Harpold and Brooks, 2018). Empirical projections of $DOS_{20}$ under the pseudo global warming scenario show that colder, drier, and sunnier sites (typical of the Rocky Mountains) are about twice as sensitive to warming as warmer, more humid, and cloudier sites (typical of the Pacific Northwest). Humid and warmer sites have relatively low snow fractions (<0.5, more rainfall effects) and thus, a smaller snowmelt signal in the diel streamflow observations. In contrast, Harpold and Brooks (2018) showed that winter ablation at SNOTEL sites in humid places, like the Pacific Northwest, are more sensitive to warming than less humid places, like the Southwest US. The difference between these findings and our streamflow-based inferences might be explained by SNOTEL sites being preferentially situated in snowy forest gaps that do not necessarily represent the catchment-scale, early-season snowmelt patterns focused on here. However, Kirchner et al. (2020) showed general agreement between SNOTEL snowmelt response and the snowmelt-induced diel streamflow signal at the warm Sagehen Creek watershed (site #23). The sensitivity of the early snowmelt timing metric ($DOS_{20}$) to climate change may be distilled into streamflow´s sensitivity to changes in precipitation partitioning (rainfall vs snowfall) and snowmelt sensitivity (more energy for melt is available); however, these two are sometimes coupled (e.g., changes in snow albedo after snowfall will alter the energy balance that controls snowmelt). Due to the empirical basis of our analysis, these two sensitivities are not easy to disentangle, but we believe that the diel analysis is better suited to investigate streamflow´s sensitivity to snowmelt changes. We focus the analysis on mostly clear-sky days, and thus implicitly exclude the effect of rainfall (or precipitation partitioning); we also use predictive variables in the MLR that relate to broad and regional snowmelt controls (i.e., seasonal meteorology) as opposed to specific

event-scale meteorology required to predict precipitation partitioning. The reliability of the empirical diel streamflow-based projections partially depends on whether climate projections are within or outside the range of observed climate conditions across the large climatic gradient found in the western US. Under the pseudo global warming scenario, cold, sunny watersheds like those in the Rocky Mountains (site #9 and #10) will shift toward more humid, warmer conditions (Figure A6), like those observed in Southern Idaho (site #29) and the northern Sierra Nevada (site #23). In contrast, the pseudo global warming scenario in places like the Pacific Northwest, particularly those involving changes in atmospheric humidity above 5 g/m$^3$ (Figure A4), have not been observed, and therefore are more uncertain. Overall, climate changes from pseudo global warming are mostly within the observed interannual and inter-watershed climate variability used to train the stepwise MLR model (Figure A4). Our empirical diel streamflow-based model implicitly assumes that other variables not included in the analysis vary together with the predictive variables (climate) and neglects variables like the catchment's physical (e.g., soil storage) and biological (e.g., vegetation) properties that do not necessarily co-vary with climate. Determining the conditions under which we can reasonably apply this type of analysis remains an open question and has been posed as one the 23 unsolved problems in hydrology (Blöschl et al., 2019), highlighting the value of comparing our empirically-based approach to a physically based model.

The sensitivity of historical snowmelt-mediated streamflow volume timing (DOQ$_{25}$ and DOQ$_{50}$) to climate change differs between the empirical diel streamflow-based approach and a land surface model, particularly in cold watersheds (Figure 9C and 9D), raising questions about current state-of-the-art projections of early season streamflow timing from NoahMP-WRF. The observed data used in the diel streamflow-based approach have larger and more variable streamflow timing responses to climate change (10 – 17 days °C$^{-1}$) in cold, dry, sunny places that are representative of small, high-elevation Rocky Mountain watersheds (Figure 8B). The historical diel streamflow analysis suggests that NoahMP-WRF may be systematically under-predicting the sensitivity of streamflow volume timing to earlier snowmelt-induced streamflow in colder and sunnier places (Figure 9C) that are most likely to have increased temperature and increased cloudiness in the future. The same mean annual future climate scenarios were applied to both approaches; however, important differences in the streamflow timing response were found between NoahMP-WRF and diel streamflow-based projections (Figure 9C and 9D). NoahMP-WRF underpredicts historical DOQ$_{25}$ (Figure 9A) across most sites, whereas the DOQ$_{50}$ is much better represented. Historically, NoahMP-WRF performed the best in rainier sites (see circled blue symbols in Figure 9A) and other sites classified as 'cloudy' and 'partly cloudy', whereas the Rocky Mountain sites, characterized by 'sunny' snowmelt event, were among the most biased (see blue filled circles in Figure 9A). This suggest that the timing of streamflow volume is better represented in areas where snowmelt processes are less important, though other variables like topographic and climatic gradients can also be important. It is worth noting that when DOQ$_{25}$ simulated by NoahMP-WRF is calculated using surface runoff only (Figure A7A) it performs better against observed DOQ$_{25}$; however, the projected sensitivity in streamflow timing to climate change remains significantly lower than predictions based on the diel-streamflow analysis (Figure A7C). The fact that NoahMP-WRF has a biased historical DOQ$_{25}$ simulation represents a challenge that goes beyond the scope of this study although these simulations have been tested

in detail in terms of the meteorology and snow components (Liu et al., 2017; Scaff et al., 2020) and have been used for climate change analyses (Musselman et al., 2017, 2018). We used these simulations in the analysis because NoahMP underlies the US National Water Model and thus its relevance to policy and research is high. There are many differences in the way that NoahMP-WRF and the empirical diel streamflow-based approach simulate the sensitivity of streamflow timing. NoahMP-WRF operating at sub-daily time steps has several advantages. For example, NoahMP-WRF can track the hourly covariance in precipitation, temperature, and humidity to estimate precipitation partitioning between rain and snow. It is also able to represent hourly radiative and turbulent energy at the snowpack, and the cold content needed to predict snowmelt. The physical hydrology is also advanced and able to consider antecedent conditions and allow evapotranspiration losses that also modulate streamflow. Despite the advantages of land surface models like NoahMP-WRF in constraining processes for future projections, the simplicity of diel streamflow-based analysis also provides several advantages. One of the main advantages is that it is derived from observations and thus it is well constrained by the observed spatial and temporal variability of snowmelt across watersheds and years (Figure 7B). Also, it does not assume anything about the complex spatial distribution of snowpacks and precipitation or subsurface properties and interactions with the surface, which are major constraints to physically-based models (Baroni et al., 2010; Christiaens and Feyen, 2001; Wilby et al., 2002). While the empirical diel streamflow-based model is not a replacement for land surface models like NoahMP-WRF, partly because the underlying streamflow datasets are not available everywhere, there is added value in including new benchmarks like the proposed $DOS_{20}$ to further constrain modeling decisions and improve model fidelity required for reliable and accurate hydrological predictions.

## 5    Conclusions

Water management in the western US relies on accurate predictions of how both short-term climate variability and long-term climate change will alter snowmelt and streamflow. Differences in predictions of snowmelt-induced streamflow between empirical diel streamflow-based projections and a land surface model (NoahMP-WRF) raise important questions about the sensitivity of streamflow timing to climate change, particularly in cold regions, and its impact on water planning. Significant differences exist in the way diel streamflow-based and land surface models predict changes to snowmelt and streamflow timing, with both approaches having strengths and weaknesses; however, the land surface model misrepresents historical patterns in streamflow response that at are more accurately estimated by the empirical model. We show that $DOS_{20}$ is a strong predictor of the early season hydrograph response, particularly in cold, sunny areas where the NoahMP-WRF streamflow timing simulations lack sensitivity to climate change. Rigorously validating future model predictions is impossible, but snowmelt and streamflow timing, inferred from diel streamflow cycles, could be used to refine land surface models and better determine the risk to valuable snow water resources (Barnett et al., 2005; Sturm et al., 2017; Viviroli et al., 2007), particularly in cold regions. Our novel approach can complement the benchmarking or calibration of physically based hydrological models, beyond typical benchmarking against daily streamflow or snow accumulation metrics. For example, the snowmelt timing metric $DOS_{20}$ based on diel streamflow observations could be used to test the performance of land surface models running at

sub-daily scales and fine spatial resolution in representing the historical snowmelt regime across watersheds and years. As land surface models move towards real application for water management (Kopp et al., 2018), the hydrology community must
seek ways to test and improve models using widely-available datasets if we are to meet the grand water management challenges posed by climate change and altered snowmelt regimes in key mountainous regions.

**Acknowledgments**

This project was supported by a grant with the Center for Weather and Water Extremes in the West (CW3E) and a National Science Foundation grant (EAR #2012310) to A.A. Harpold. S.A. Krogh thanks CONICYT for providing financial support
through the Becas Chile program for postdoctoral studies. We appreciate the positive feedback and suggestions from Professor Jessica Lundquist and an anonymous reviewer, which greatly improved the paper.

**Contributions**

SK and AH designed the study. SK performed all the analyses, prepared the figures, and drafted the first version of the
manuscript. JK developed the 'diel cycle index' which served as the initial idea for the presented snowmelt detection method. GS collected and pre-processed USGS hourly streamflow data and NLDAS-2 solar radiation. LS pre-processed daily surface and subsurface runoff from the WRF CONUS-I simulations. All the authors reviewed and contributed to the final version of the manuscript.

**Competing Interests**
The authors declare that they have no conflict of interest.

**Code/Data availability**
Data from NoahMP-WRF simulations can be access through their public website https://rda.ucar.edu/datasets/ds612.5/. Hourly
shortwave radiation can be accessed online through: https://ldas.gsfc.nasa.gov/nldas/v2/forcing. Hourly streamflow from the USGS database can be accessed online through: https://waterdata.usgs.gov/nwis/sw. The code used to process and analyze the data presented in the study is available upon request to the corresponding author.

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

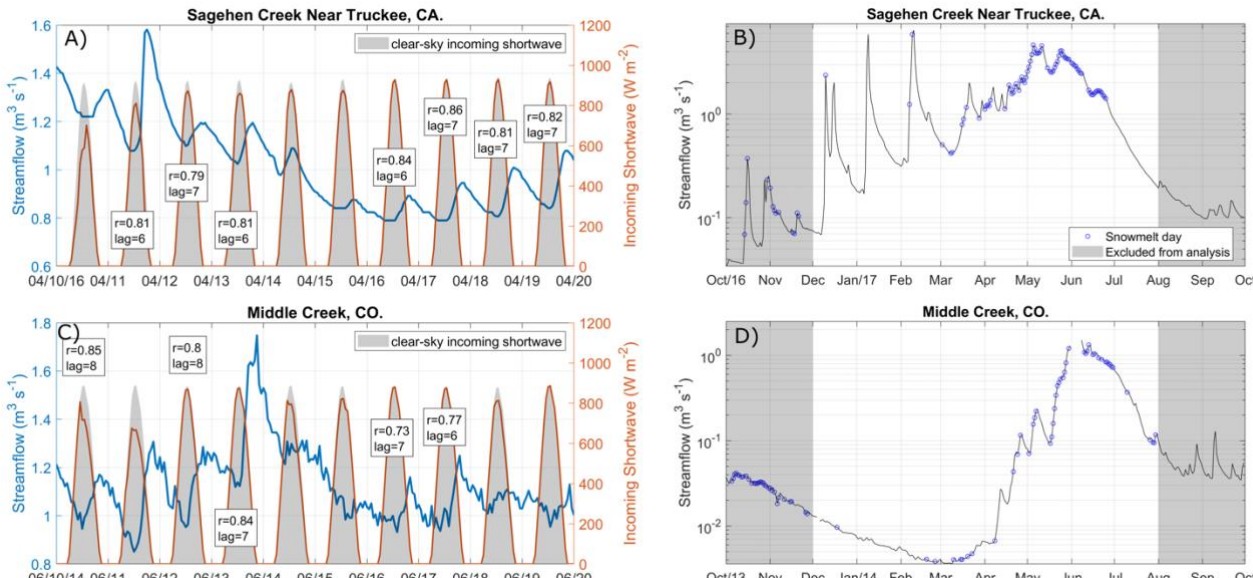

**Figure 1: Examples of the diel cycle analysis applied to two watersheds located in California (A) (B) (WY2016) and Colorado (C) (D) (WY2014). (A) and (C) show hourly solar radiation (orange) and streamflow (blue); the first statistically significant (p<0.01) lagged spearman correlation (r>0.6) between streamflow and solar radiation is shown on a text box for clear-sky days only (>80% of clear-sky solar radiation). (B) and (D) show the solar radiation-driven snowmelt days (blue circles) on top of the annual hydrograph (semi-log scale) for the period of analysis (white background, December to July).**

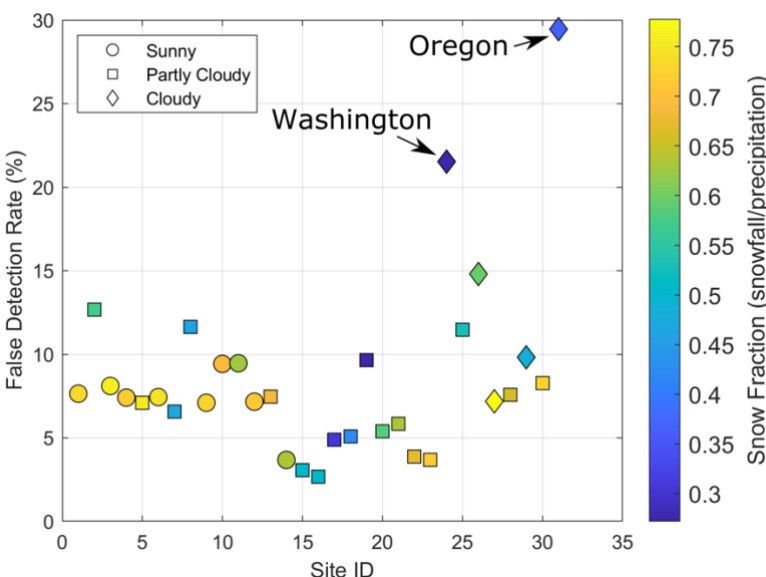

**Figure 2: Percentage of days that were classified as having snowmelt following the diel streamflow cycle analysis that also had daily precipitation above 5 mm and a mean daily air temperature above 2 ᵒC. Symbols are associated with the mean annual percentage of snowmelt days under clear-sky conditions. Sunny sites (circles) have >90%, clear-sky snowmelt days, partly cloudy sites (squares) have between 70 and 90%, and cloudy sites (diamonds) have <70% clear-sky snowmelt days. Clear-sky snowmelt days are defined as those with more than 80% of the potential clear-sky solar radiation.**

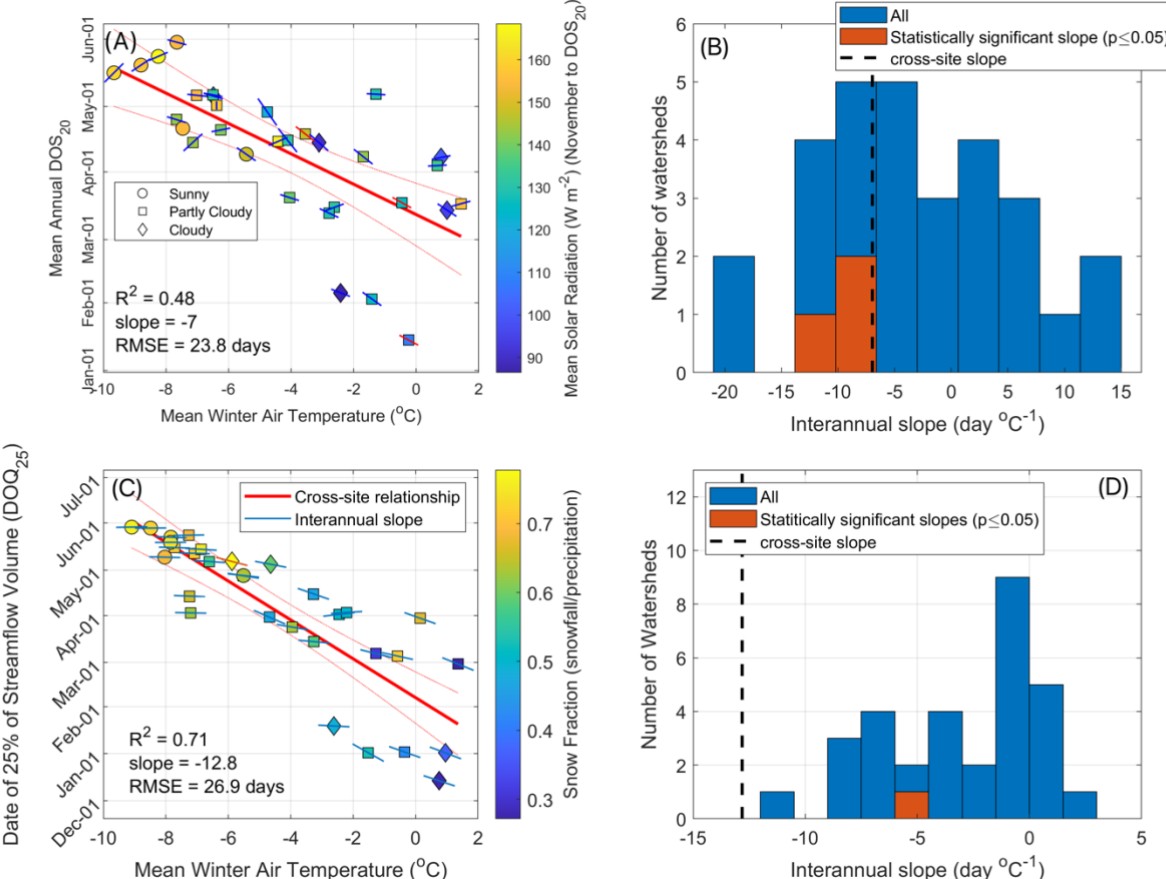

**Figure 3:** (A) and (C) show cross-site relationships between mean winter air temperature (November to February) and DOS$_{20}$ and the date of 25% of annual streamflow volume (DOQ$_{25}$), respectively. Slopes of individual sites' interannual relationships are shown as the lines on top of each symbol, where statistically significant (p-value ≤0.05) slopes are red. Non-significant interannual slopes are presented to show the overall tendency in their spatial distribution. Symbols are associated with the mean annual percentage of snowmelt days under clear-sky conditions. Sunny sites (circles) have >90% clear-sky snowmelt days, partly cloudy sites (squares) have between 70 and 90%, and cloudy sites (diamonds) have <70% clear-sky snowmelt days. Clear-sky snowmelt days are defined as those with more than 80% of the potential clear-sky solar radiation. (B) and (D) show histograms of interannual slopes (for all watershed and those with statistically significant relationships) and the cross-site relationships presented in their respective left panel.

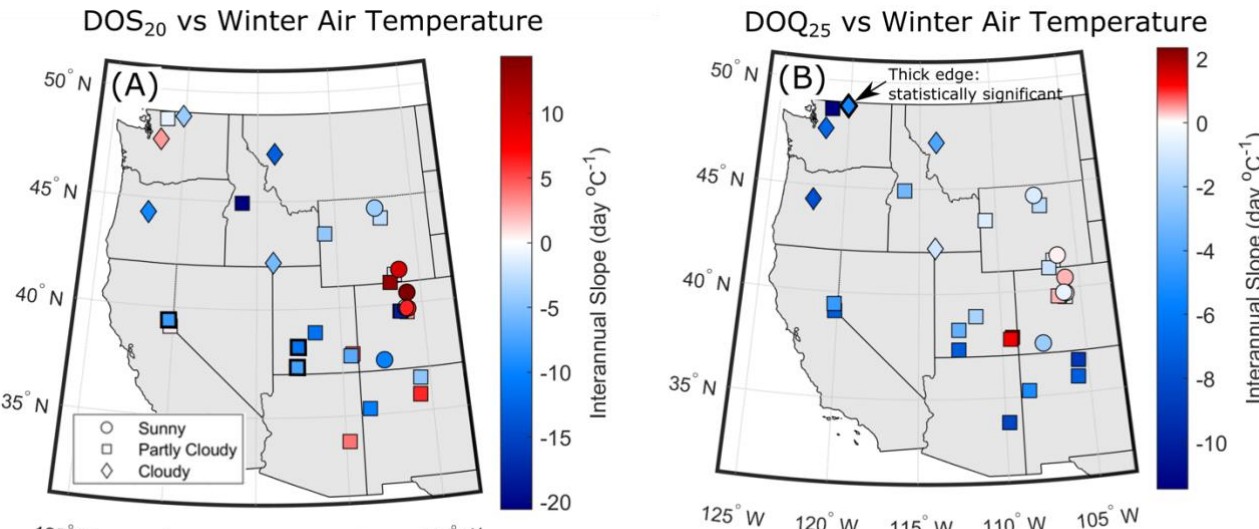

**Figure 4: Spatial variability of watershed-level interannual slopes for (A) $DOS_{20}$ vs winter air temperature, and (B) $DOQ_{25}$ vs winter air temperature. Watersheds with statistically significant relationships are highlighted in symbols with thicker edges and are associated with those presented in Figure 3. Symbols are associated with the mean annual percentage of snowmelt days under clear-sky conditions. Sunny sites (circles) have >90% clear-sky snowmelt days, partly cloudy sites (squares) have between 70 and 90%, and cloudy sites (diamonds) have <70% clear-sky snowmelt days. Clear-sky snowmelt days are defined as those with more than 80% of the potential clear-sky solar radiation.**

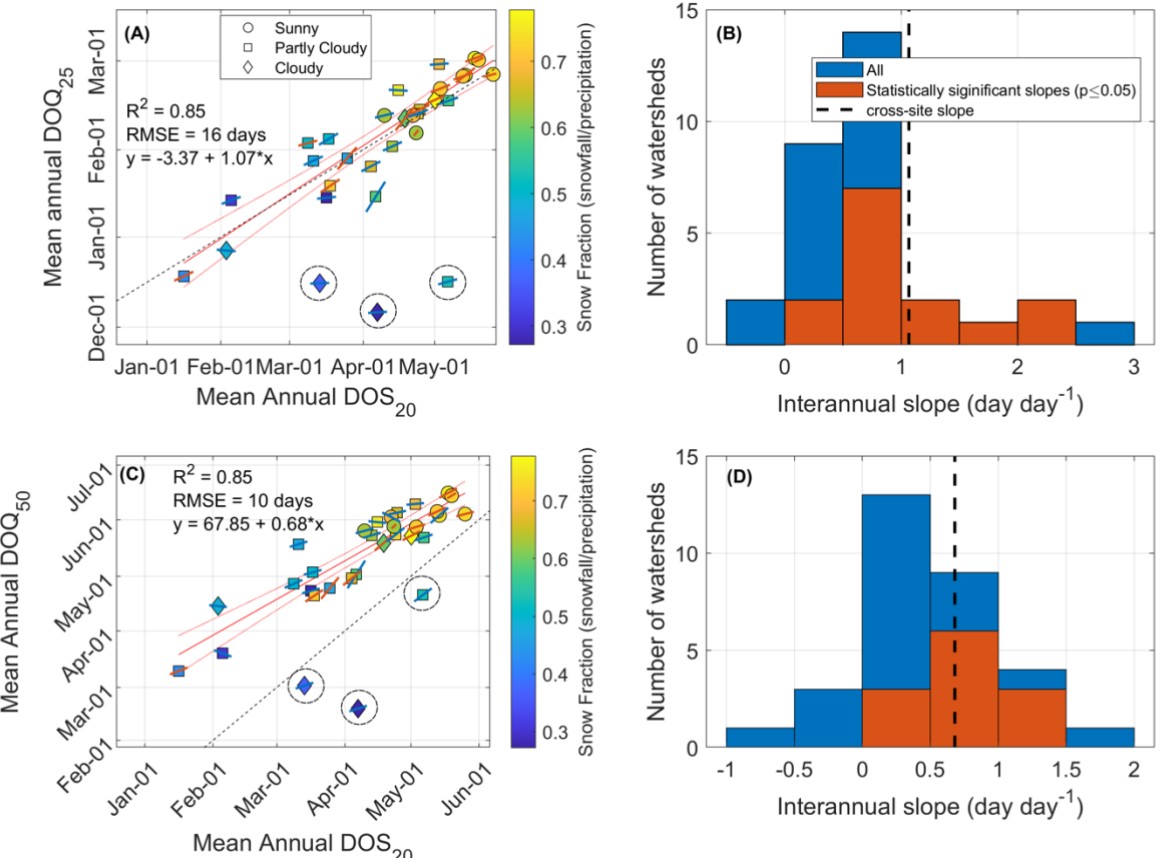

**Figure 5: (A)** The day when the 20th percentile of snowmelt days occurs (DOS$_{20}$), compared to the date of the 25% of the annual streamflow volume (DOQ$_{25}$). **(C)** DOS$_{20}$ against the date of 50% of the annual streamflow volume (DOQ$_{50}$). Dashed lines in (A) and (C) are 1:1 lines, and the slopes of sites' interannual relationships are shown as the lines on top of each symbol, with statistically significant (p-value ≤0.05) slopes shown in red. Sites #24, #25 and #31, indicated by dashed circles, fall far from the linear regression and are not included in its calculation. Symbols indicate the mean annual percentage of clear-sky snowmelt days, where sunny sites (circles) have >90% clear-sky snowmelt days, partly cloudy sites (squares) have between 70 and 90%, and cloudy sites (diamonds) have <70%; clear-sky snowmelt days are defined as those with more than 80% of the potential clear-sky solar radiation. **(B)** and **(D)** show histograms of interannual slopes (for all watershed and those with statistically significant relationships) and the cross-site relationships presented in their respective left panels.

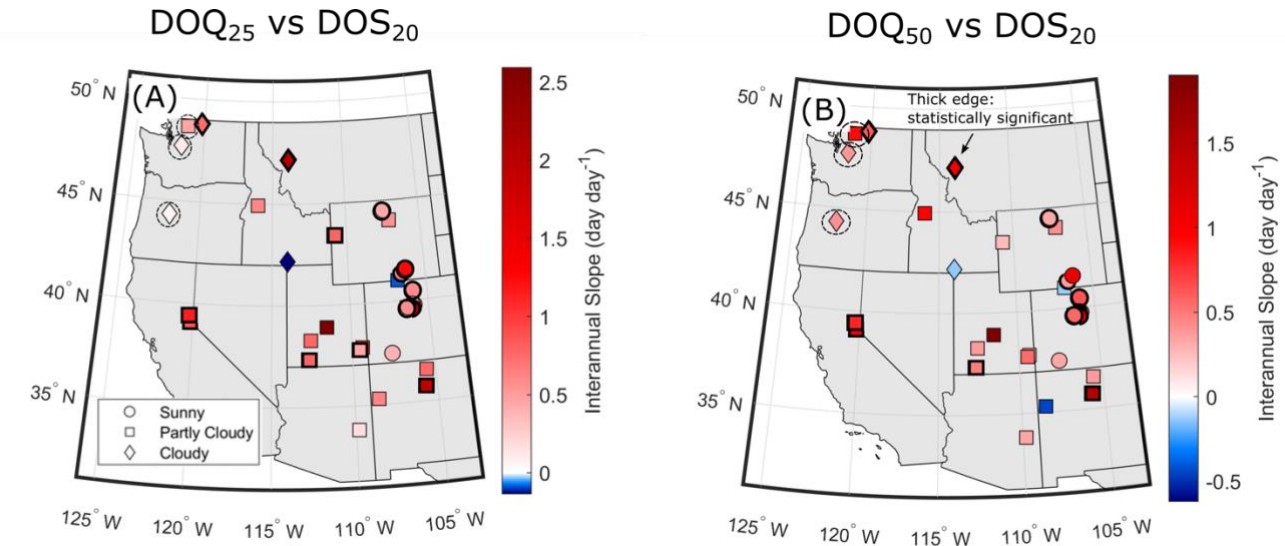

**Figure 6: Spatial variability of the watershed-level interannual slopes for (A) DOQ$_{25}$ vs DOS$_{20}$, and (B) DOQ$_{50}$ vs DOS$_{20}$. Watersheds with statistically significant relationships are highlighted in symbols with thicker edges and are associated with those presented in Figure 5. Watersheds that fall far from the linear regression presented in Figure 5 are surrounded by a dashed circle. Symbols are associated with the mean annual percentage of snowmelt days under clear-sky conditions. Sunny sites (circles) have >90% clear-sky snowmelt days, partly cloudy sites (squares) have between 70 and 90%, and cloudy sites (diamonds) have <70%. Clear-sky snowmelt days are defined as those with more than 80% of the potential clear-sky solar radiation.**

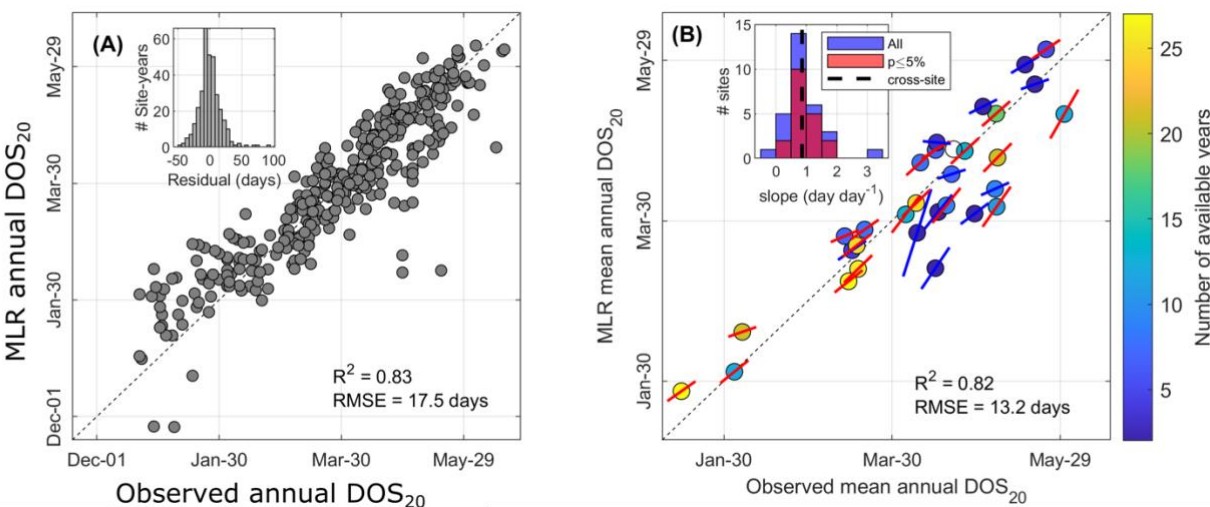

**Figure 7: (A)** Scatterplot showing the fit of the stepwise multiple linear regression (MLR) model to the observed $DOS_{20}$ across all sites and years. **(B)** shows the same stepwise MLR model applied at the mean annual watershed level across all watersheds. Interannual variability represented by the slope of the linear relationship is shown as a line overlapping each circle (i.e., watershed); red and blue lines indicate statistically significant ($p \leq 0.05$) and insignificant slopes, respectively.

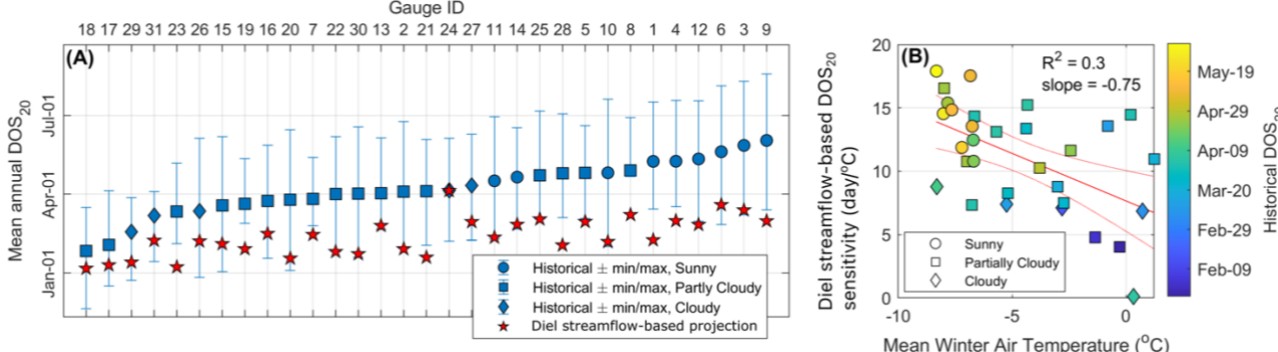

**Figure 8: (A) Historical DOS$_{20}$ from the diel analysis and projected changes in DOS$_{20}$ using the empirical diel streamflow-based projections under the RCP 8.5 pseudo global warming climate for the end of the 21st century. Watersheds are sorted from earlier (left) to later (right) historical DOS$_{20}$. Symbols associated with future projections (stars) are not classified by sunny, partly cloudy, or cloudy, as we make no inference about the cloudiness condition of snowmelt days under the climate change scenario. Blue symbols in (A) represent the mean annual percentage of clear-sky snowmelt days, where sunny sites (circles) have >90% clear-sky snowmelt days, partly cloudy sites (squares) have between 70 and 90%, and cloudy sites (diamonds) have <70%. Clear-sky snowmelt days are defined as those with more than 80% of the potential clear-sky solar radiation. (B) Relationship between mean winter air temperature and the sensitivity of DOS$_{20}$ to climate change as projected by the empirical diel streamflow-based model.**

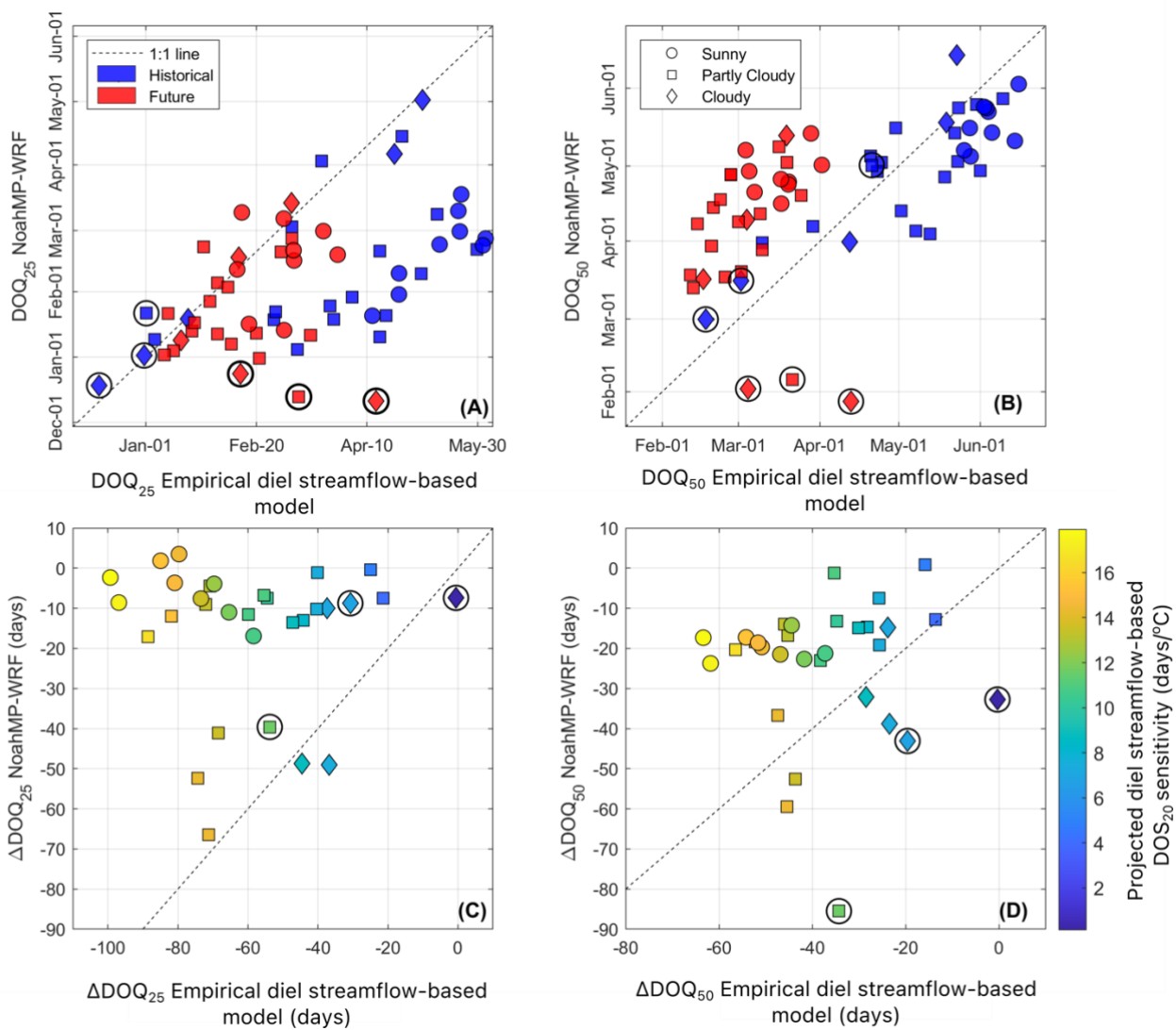

**Figure 9: Changes to DOQ$_{25}$ and DOQ$_{50}$ due to climate change under an RCP8.5 pseudo global warming climate scenario by the end of the century. (A) and (B) compare historical against projected values between NoahMP-WRF and the empirical diel streamflow-based model. (C) and (D) compare the projected change in streamflow timing (future minus historical) between NoahMP-WRF and the empirical diel streamflow-based model, colored by the sensitivity of DOS$_{20}$ to climate change as projected by the empirical diel streamflow-based model (Figure 8b). Symbols surrounded by black circles indicate sites that were excluded from the regression analysis in Figure 5 (rainier sites #24, #25 and #31). Symbols represent the historical mean annual percentage of clear-sky snowmelt days, where sunny sites (circles) have >90% clear-sky snowmelt days, partly cloudy sites (squares) have between 70 and 90%, and cloudy sites (diamonds) have <70%; clear-sky snowmelt days are defined as those with more than 80% of the potential clear-sky solar radiation. We make no inference about the cloudiness condition of snowmelt days under the RCP8.5 P climate scenario; however, red symbols (upper panels) follow the same symbology for easier interpretation.**

**Table 1: List of Abbreviations**

| Abbreviation | Definition |
|---|---|
| CAMELS | Catchments Attributes and MEteorology for Large-sample Studies |
| $DOQ_{25}$ | Date of 25% of annual streamflow volume |
| $DOQ_{50}$ | Date of 50% of annual streamflow volume |
| $DOS_{20}$ | The day when the 20th percentile of the snowmelt days occurs, with snowmelt days as defined by the streamflow diel cycle analysis |
| GCM | Global Climate Model |
| MLR | Multiple Linear Regression Model |
| NLDAS-2 | Phase 2 of the National Land Data Assimilation System |
| Noah-MP | Noah Multi Parameterization land surface model |
| NoahMP-WRF | Simulations by WRF using the Noah-MP land surface model |
| RCP8.5 | Representative Concentration Pathway 8.5 |
| WRF | Weather Research and Forecasting Model |

**Table 2: List of the 31 watersheds from the CAMELS dataset included in this study. Data from Addor et al. (2017).**

| ID | USGS ID | Watershed Name | Drainage Area (km²) | Mean Elevation (masl) | Mean slope (m km⁻¹) | Lat. (°N) | Lon. (°W) | Snow Fraction | Aridity index | Soil Depth (m) |
|---|---|---|---|---|---|---|---|---|---|---|
| 1 | 06278300 | Shell Creek, WY. | 58.9 | 2,953 | 86.7 | 44.51 | 107.40 | 0.73 | 1.32 | 0.74 |
| 2 | 06311000 | North Fork Powder River, WY. | 61.2 | 2,516 | 41.1 | 44.03 | 107.08 | 0.57 | 1.68 | 0.90 |
| 3 | 06614800 | Michigan River, CO. | 4.0 | 3,297 | 145.8 | 40.50 | 105.87 | 0.76 | 1.29 | 0.57 |
| 4 | 06622700 | North Brush Creek, WY. | 98.7 | 2,837 | 71.3 | 41.37 | 106.52 | 0.72 | 1.48 | 2.20 |
| 5 | 06623800 | Encampment River, WY. | 187.7 | 2,971 | 90.9 | 41.02 | 106.82 | 0.75 | 1.06 | 1.14 |
| 6 | 06632400 | Rock Creek, WY. | 163.0 | 3,002 | 69.0 | 41.59 | 106.22 | 0.74 | 1.46 | 2.52 |
| 7 | 08267500 | Rio Hondo, NM. | 96.3 | 3,007 | 149.1 | 36.54 | 105.56 | 0.47 | 2.12 | 0.50 |
| 8 | 08377900 | Rio Mora, NM. | 139.0 | 3,018 | 105.3 | 35.78 | 105.66 | 0.47 | 1.50 | 0.85 |
| 9 | 09034900 | Bobtail Creek, CO. | 15.7 | 3,571 | 102.8 | 39.76 | 105.91 | 0.73 | 1.16 | 0.47 |
| 10 | 09035900 | South Fork of Williams Fork, CO. | 72.8 | 3,241 | 123.9 | 39.80 | 106.03 | 0.69 | 1.44 | 0.56 |
| 11 | 09047700 | Keystone Gulch, CO. | 23.6 | 3,334 | 103.8 | 39.59 | 105.97 | 0.63 | 1.92 | 0.45 |
| 12 | 09066200 | Booth Creek, CO. | 16.1 | 3,072 | 145.4 | 39.65 | 106.32 | 0.71 | 1.40 | 0.27 |
| 13 | 09066300 | Middle Creek, CO. | 15.5 | 2,944 | 143.8 | 39.65 | 106.38 | 0.69 | 1.49 | 0.48 |
| 14 | 09352900 | Vallecito Creek, CO. | 188.2 | 3,283 | 156.1 | 37.48 | 107.54 | 0.63 | 1.24 | 0.50 |
| 15 | 09378170 | South Creek, UT. | 21.9 | 2,308 | 67.7 | 37.85 | 109.37 | 0.50 | 1.79 | 1.16 |
| 16 | 09378630 | Recapture Creek, UT. | 10.4 | 2,125 | 53.4 | 37.76 | 109.48 | 0.50 | 1.88 | 0.55 |
| 17 | 09386900 | Rio Nutria, NM. | 184.9 | 2,342 | 37.4 | 35.28 | 108.55 | 0.31 | 2.48 | 1.07 |
| 18 | 09404450 | East Fork Virgin River, UT. | 193.0 | 2,070 | 56.2 | 37.34 | 112.60 | 0.42 | 2.86 | 0.82 |
| 19 | 09492400 | East Fork White River, AZ. | 129.0 | 2,469 | 65.4 | 33.82 | 109.81 | 0.27 | 1.88 | 0.92 |
| 20 | 10205030 | Salina Creek, UT. | 134.6 | 2,489 | 76.2 | 38.91 | 111.53 | 0.58 | 2.46 | 0.67 |
| 21 | 10234500 | Beaver River, UT. | 236.4 | 2,499 | 95.2 | 38.28 | 112.57 | 0.63 | 2.06 | 0.60 |

| 22 | 10336660 | Blackwood Creek, CA. | 29.8 | 2,113 | 83.5 | 39.11 | 120.16 | 0.67 | 0.77 | 0.79 |
|----|----------|----------------------|------|-------|------|-------|--------|------|------|------|
| 23 | 10343500 | Sagehen Creek, CA. | 27.6 | 2,157 | 81.2 | 39.43 | 120.24 | 0.71 | 1.10 | 1.20 |
| 24 | 12147600 | South Fork Tolt River, WA. | 14.1 | 1,068 | 159.4 | 47.71 | 121.60 | 0.27 | 0.22 | 0.63 |
| 25 | 12178100 | Newhalem Creek, WA. | 69.7 | 1,305 | 255.7 | 48.66 | 121.24 | 0.53 | 0.33 | 0.54 |
| 26 | 12381400 | South Fork Jocko River, MT. | 151.0 | 1,877 | 102.2 | 47.20 | 113.85 | 0.59 | 0.97 | 0.62 |
| 27 | 12447390 | Andrews Creek, WA. | 58.1 | 1,701 | 172.6 | 48.82 | 120.15 | 0.78 | 0.86 | 0.47 |
| 28 | 13018300 | Cache Creek, WY. | 27.9 | 2,198 | 109.5 | 43.45 | 110.70 | 0.66 | 1.50 | 0.69 |
| 29 | 13083000 | Trapper Creek, ID. | 133.2 | 1,863 | 69.1 | 42.17 | 113.98 | 0.49 | 2.11 | 1.04 |
| 30 | 13240000 | Lake Fork Payette River, ID. | 125.6 | 1,965 | 110.1 | 44.91 | 116.00 | 0.73 | 0.75 | 0.44 |
| 31 | 14158790 | Smith River, OR. | 40.6 | 1,027 | 116.4 | 44.33 | 122.05 | 0.37 | 0.36 | 0.85 |

# 7    Appendices

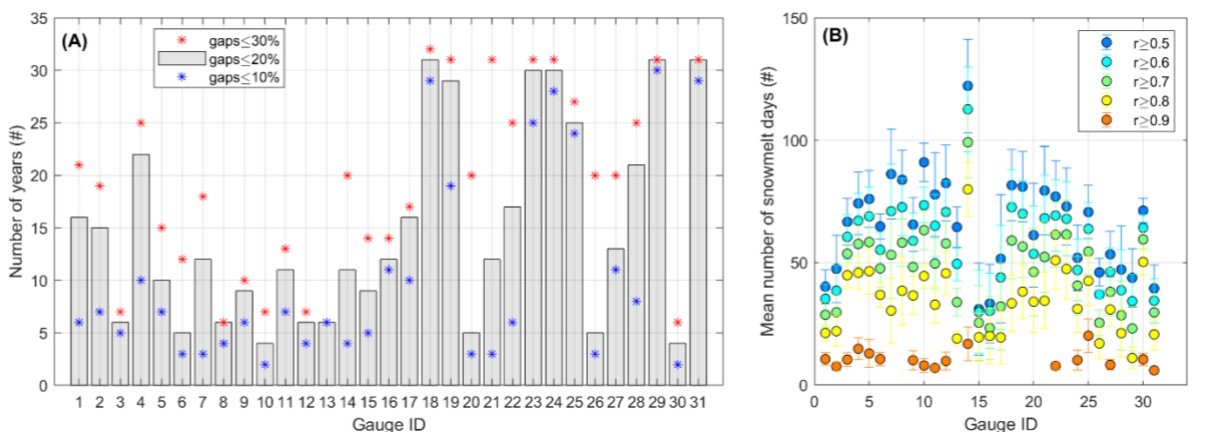

**Figure A1: (A)** Number of available years with less than 30, 20 and 10% gaps in days with hourly streamflow records between December 1 and August 1. Gauge ID is as presented in Table 2. Numbers of years at site #13 are the same for all thresholds (overlapping symbols). **(B)** Sensitivity of the mean annual number of detected snowmelt days to different Spearman correlation cutoffs (0.5, 0.6, 0.7 and 0.9) between hourly solar radiation and streamflow. Error bar represents the standard deviation.

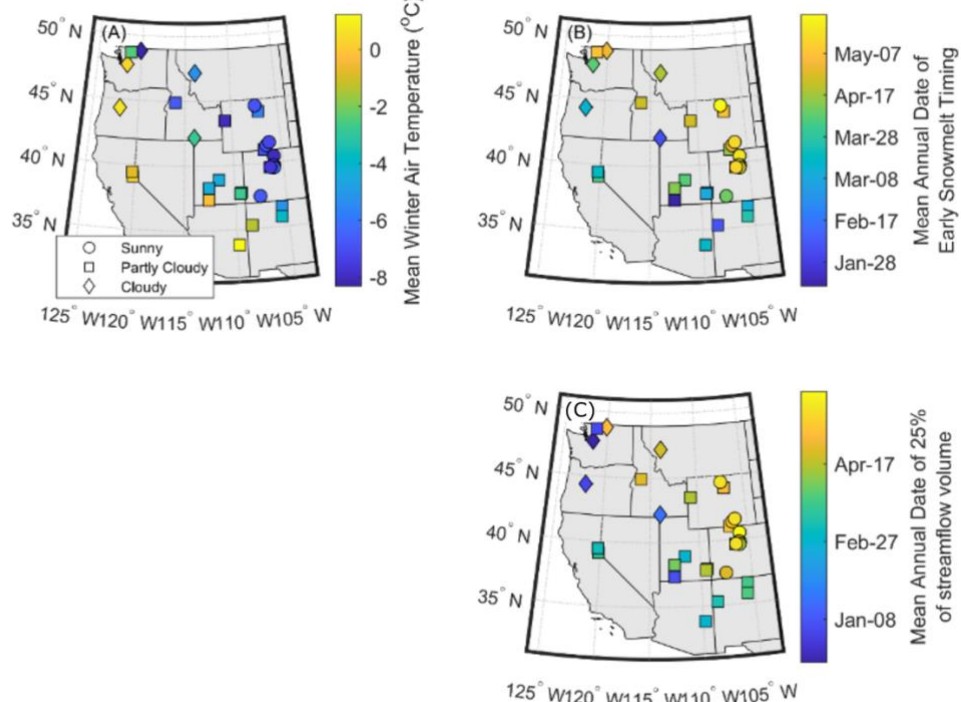

**Figure A2: (A): CAMELS mean winter (November to February) air temperature, (B) mean annual DOS$_{20}$, and (C) mean annual DOQ$_{25}$. Symbols (circle, square and diamond) represent the mean annual percentage of clear-sky snowmelt days, where sunny sites have >90% clear-sky snowmelt days, partly cloudy have between 70 and 90%, and cloudy have <70%; clear-sky snowmelt days are defined as those with more than 80% of the potential clear-sky solar radiation.**

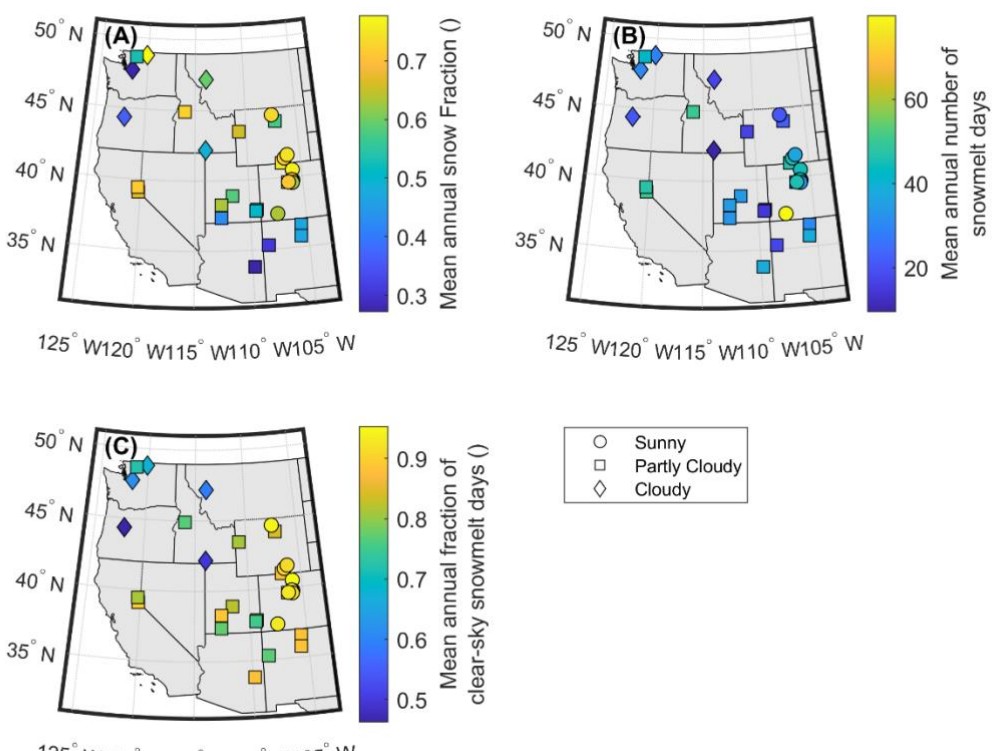

**Figure A3: (A): CAMELS mean annual snow fraction (snowfall/precipitation), (B) mean annual number of snowmelt days between December 1 and August 1 (calculated as the days with a correlation between hourly solar radiation and lagged streamflow greater than 0.8), and (C) mean annual fraction of clear-sky snowmelt days, calculated as the number of snowmelt days with clear-sky conditions as a fraction of total snowmelt days. A clear-sky snowmelt day is defined as having more than 80% of the potential clear-sky solar radiation. Symbols (circle, square and diamond) represent the mean annual percentage of clear-sky snowmelt days, where sunny sites have >90% clear-sky snowmelt days, partly cloudy have between 70 and 90%, and cloudy have <70.**

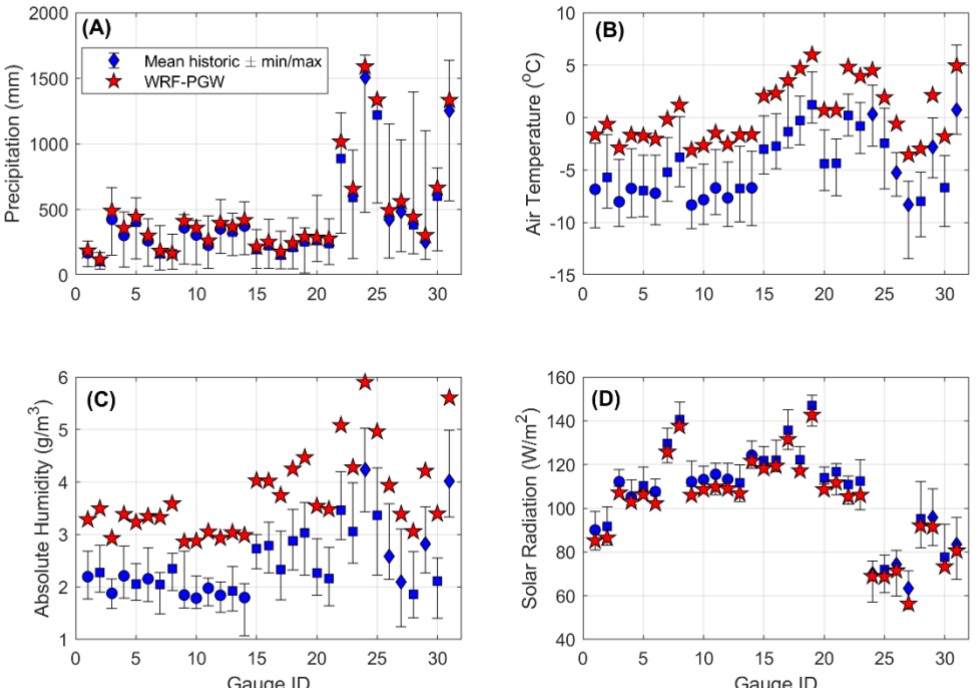

**Figure A4: Historic winter climate variability for each predictor used in the stepwise MLR model (Equation 1) for the period between November and $DOS_{20}$ in blue. (A) Precipitation, (B) air temperature, (C) absolute humidity and (D) solar radiation. In red are the perturbed mean climate variables under the RCP8.5 pseudo global warming scenario by the end of the century. This analysis suggests that most of the climate change signal from NoahMP-WRF pseudo global warming is within the observed climate variability, except for air temperature and atmospheric humidity in some watersheds. Blue symbols (circle, square and diamond) associated with historical values represent the mean annual percentage of clear-sky snowmelt days, where sunny sites have >90% clear-sky snowmelt days, partly cloudy have between 70 and 90%, and cloudy have <70%; clear-sky snowmelt days are defined as those with more than 80% of the potential clear-sky solar radiation. We make no inference about the cloudiness condition of snowmelt days under the RCP8.5 pseudo global warming scenario, and thus, we use a five-point star (in red) for the future scenario.**

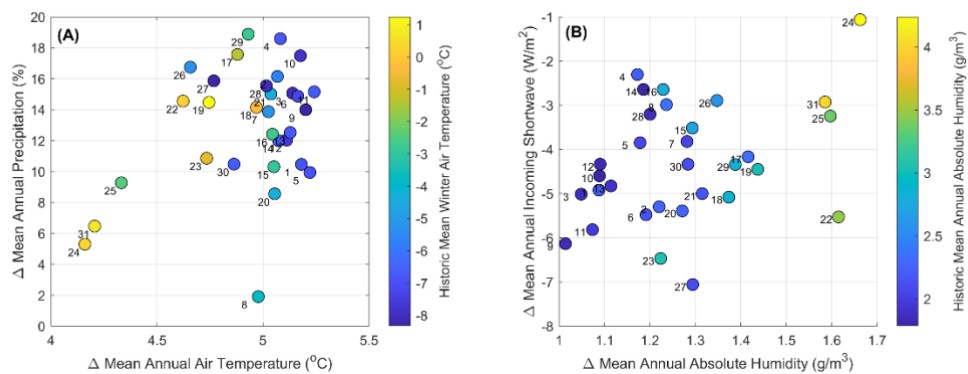

**Figure A5: Mean annual climate changes projected by WRF under an RCP8.5 pseudo global warming scenario by the end of the century. (A) shows changes in precipitation against air temperature. (B) shows incoming shortwave against absolute humidity. Numbers represent the Gauge IDs as presented in Table 2.**

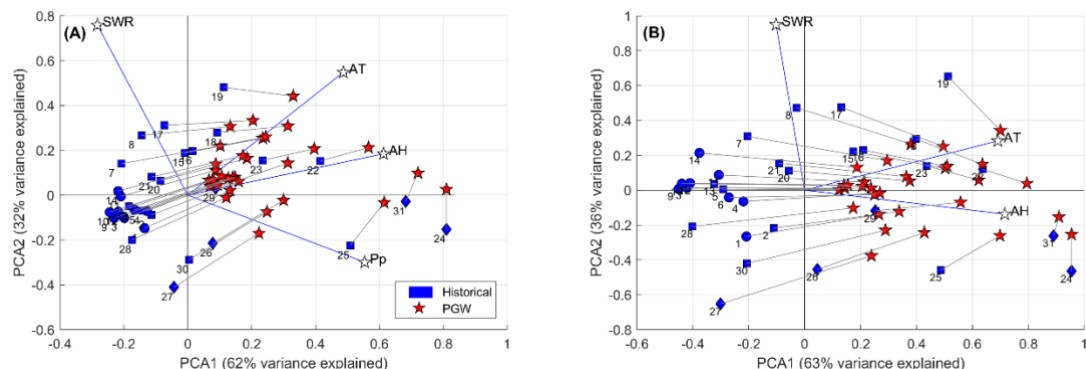

**Figure A6: (A) Principal Component Analysis for historical precipitation (Pp), air temperature (AT), absolute humidity (AH) and shortwave radiation (SWR) at each watershed, and the changes associated with the pseudo global warming as simulated by WRF. (B) shows the same analysis but excluding precipitation from the analysis. Blue symbols (circle, square and diamond) associated with historical values represent the mean annual percentage of clear-sky snowmelt days, where sunny sites have >90% clear-sky snowmelt days, partly cloudy have between 70 and 90%, and cloudy have <70%; clear-sky snowmelt days are defined as those with more than 80% of the potential clear-sky solar radiation. We make no inference about the cloudiness condition during snowmelt days under the RCP8.5 pseudo global warming scenario, and thus, we use a five-point star (in red) for the future scenario. Numbers next to blue symbols represent the Gauge IDs as presented in Table 2.**

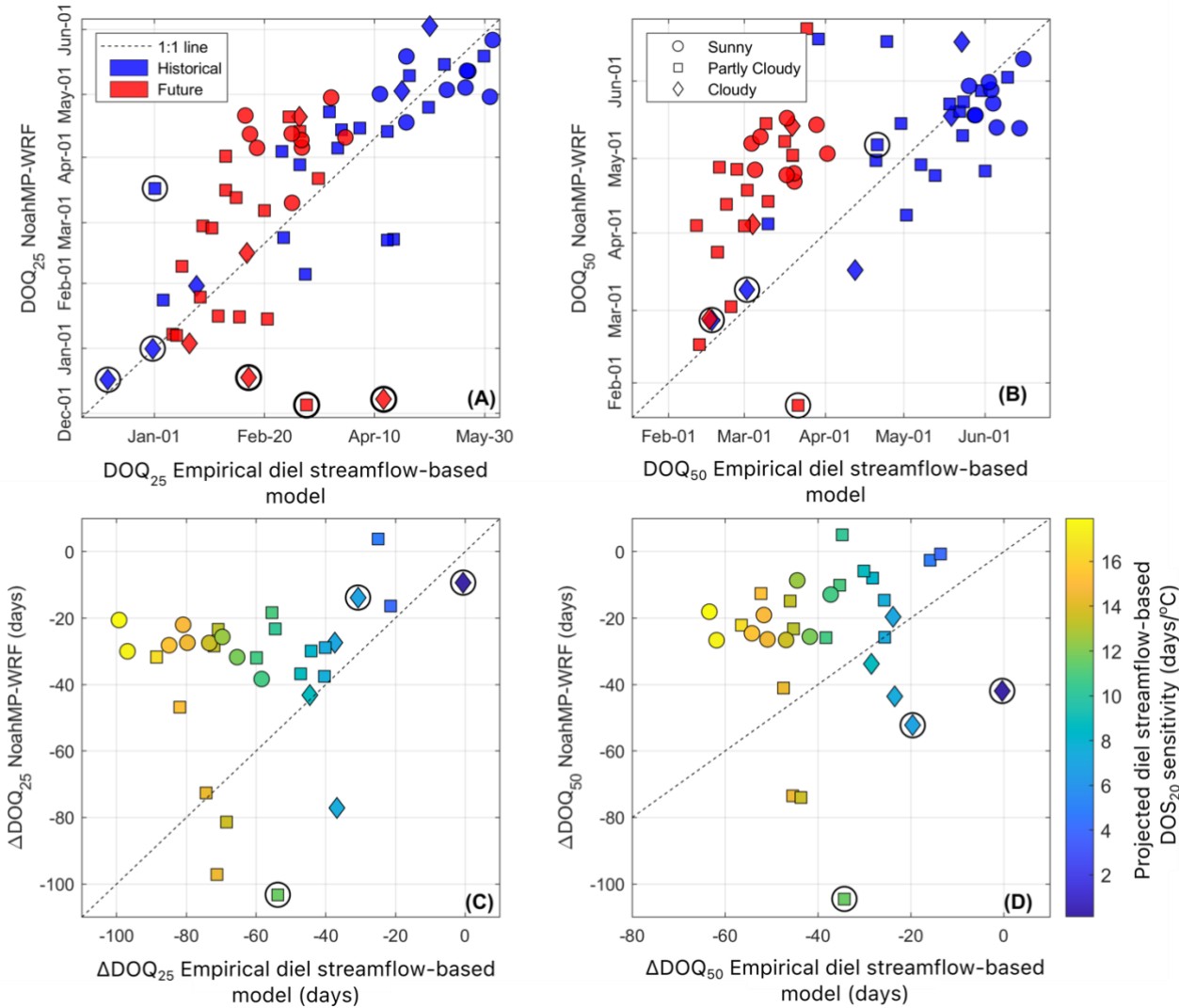

**Figure A7: Same as Figure 9 but using streamflow timing metrics from NoahMP-WRF for an RCP8.5 pseudo global warming**
**scenario, calculated using surface runoff only instead of using surface plus subsurface runoff (as in Figure 6). Note the improved fit in historical DOQ$_{25}$; however, this analysis yields very similar results to those of Figure 6, with NoahMP-WRF streamflow simulations being much less sensitive to climate change than the empirical diel streamflow-based model suggests. (A) and (B) compare historical against projected values between NoahMP-WRF and the empirical diel streamflow-based model. (C) and (D) compare the projected change (future minus historical) between NoahMP-WRF and the diel streamflow-based model, colored by**
**the sensitivity of DOS$_{20}$ to climate change as projected by the empirical diel streamflow-based model (Figure 5b). Symbols surrounded by black circles indicate sites that were excluded from the regression analysis in Figure 3 (rainier sites #24, #25 and #31). Symbols (circle, square and diamond) represent the historical mean annual percentage of clear-sky snowmelt days, where sunny sites have >90% clear-sky snowmelt days, partly cloudy have between 70 and 90%, and cloudy have <70%; clear-sky snowmelt days are defined as those with more than 80% of the potential clear-sky solar radiation. We make no inference about the**
**cloudiness condition of snowmelt days under the RCP8.5 pseudo global warming climate scenario; however, red symbols (upper panels) follow the same symbology for easier interpretation.**

Table A1: Coefficient of determination ($R^2$) and slope (in parenthesis, day/day) of the linear regression between different early
snowmelt timing metrics and $DOQ_{25}$ and $DOQ_{50}$, as presented in Figure 5, for different correlation cutoffs (r) between hourly solar radiation and streamflow. DOSxx represent the date when the xx[th] percentile of snowmelt days occurs. Sites #24, #35 and #31, are excluded from the linear relationship. Bolded numbers are those used in the result and discussion sections.

| Early snowmelt timing metrics | | vs $DOQ_{25}$ | vs $DOQ_{50}$ |
|---|---|---|---|
| r > 0.5 | 1st snowmelt day | 0.13 (0.61) | 0.06 (0.25) |
| | 1st 3 consecutive snowmelt day | 0.5 (0.71) | 0.4 (0.4) |
| | $DOS_5$ | 0.37 (0.83) | 0.28 (0.45) |
| | $DOS_{10}$ | 0.49 (0.91) | 0.43 (0.52) |
| | $DOS_{20}$ | 0.69 (1.1) | 0.66 (0.67) |
| | $DOS_{30}$ | 0.73 (1.1) | 0.72 (0.68) |
| r > 0.6 | 1st snowmelt day | 0.24 (0.73) | 0.15 (0.35) |
| | 1st 3 consecutive snowmelt day | 0.59 (0.77) | 0.49 (0.44) |
| | $DOS_5$ | 0.46 (0.82) | 0.37 (0.45) |
| | $DOS_{10}$ | 0.63 (0.97) | 0.53 (0.55) |
| | $DOS_{20}$ | 0.76 (1.05) | 0.72 (0.64) |
| | $DOS_{30}$ | 0.77 (1.07) | 0.78 (0.67) |
| r > 0.7 | 1st snowmelt day | 0.42 (0.73) | 0.3 (0.39) |
| | 1st 3 consecutive snowmelt day | 0.62 (0.85) | 0.59 (0.53) |
| | $DOS_5$ | 0.61 (0.86) | 0.51 (0.49) |
| | $DOS_{10}$ | 0.71 (0.94) | 0.63 (0.55) |
| | $DOS_{20}$ | 0.76 (0.99) | 0.75 (0.62) |
| | $DOS_{30}$ | 0.79 (1.03) | 0.82 (0.65) |
| r > 0.8 | 1st snowmelt day | 0.66 (0.87) | 0.54 (0.5) |
| | 1st 3 consecutive snowmelt day | 0.76 (1.09) | 0.78 (0.71) |
| | $DOS_5$ | 0.79 (1.01) | 0.7 (0.6) |
| | $DOS_{10}$ | 0.83 (1.03) | 0.78 (0.64) |
| | **$DOS_{20}$** | **0.85 (1.07)** | **0.85 (0.68)** |
| | $DOS_{30}$ | 0.85 (1.1) | 0.88 (0.72) |

**Table A2: Root mean square error (RMSE) and coefficient of determination ($R^2$, in parentheses) associated with several stepwise multiple linear regressions (similar to the one in Equation 1) using different early snowmelt timing metrics (e.g., Equation 1 uses $DOS_{20}$) and correlation cutoffs (r) between hourly solar radiation and streamflow used to define snowmelt days. DOSxx represents the date when the xx[th] percentile of snowmelt days occurs. Bolded numbers are associated with the stepwise MLR in Equation 1 also shown in Figure 7A.**

| Early snowmelt timing metrics | r > 0.5 | r > 0.6 | r > 0.7 | r > 0.8 |
|---|---|---|---|---|
| First snowmelt day | 11.1 (0.87) | 12.3 (0.88) | 15.2 (0.88) | 21.7 (0.82) |
| First 3 consecutive snowmelt days | 24.6 (0.8) | 24.8 (0.8) | 26.1 (0.77) | 20.2 (0.8) |
| $DOS_5$ | 14.9 (0.83) | 15.4 (0.85) | 17.3 (0.86) | 21.1 (0.8) |
| $DOS_{10}$ | 16.4 (0.82) | 17.3 (0.83) | 19.9 (0.82) | 19.6 (0.82) |
| $DOS_{20}$ | 16.5 (0.82) | 17.9 (0.82) | 18.9 (0.82) | **17.5 (0.83)** |
| $DOS_{30}$ | 16.3 (0.82) | 17.4 (0.82) | 17.8 (0.82) | 16.3 (0.83) |

**Table A3: Coefficient of determination ($R^2$) for the site-average stepwise multiple linear regression, analogous to that presented in Figure 7B, for different modeling decisions (correlation cutoff between hourly solar radiation and streamflow, r, and early snowmelt days metrics). DOSxx represents the date when the xx[th] percentile of snowmelt days occurs. Bolded number is associated with the stepwise MLR in Equation 1 using DOS$_{20}$.**

| Early snowmelt timing metrics | r > 0.5 | r > 0.6 | r > 0.7 | r > 0.8 |
|---|---|---|---|---|
| First snowmelt day | 0.8 | 0.82 | 0.89 | 0.79 |
| First 3 consecutive snowmelt days | 0.81 | 0.77 | 0.73 | 0.69 |
| DOS$_5$ | 0.84 | 0.85 | 0.87 | 0.83 |
| DOS$_{10}$ | 0.84 | 0.85 | 0.86 | 0.84 |
| DOS$_{20}$ | 0.83 | 0.82 | 0.82 | **0.82** |
| DOS$_{30}$ | 0.83 | 0.81 | 0.81 | 0.8 |

Table A4: Standardized beta coefficients for the stepwise MLR associated with the different correlation cutoffs (r) between hourly solar radiation and streamflow, and different early snowmelt metrics. These stepwise MLR models follow the same structure as that of Equation 1; however, in this case predictors were standardized to estimate their relative importance. AT: Air Temperature, Pp: Precipitation, RH: Relative Humidity, SWR: Incoming Shortwave Radiation. DOSxx represent the date when the $xx^{th}$ percentile of snowmelt days occurs. *Indicates rows that do not meet all the MLR assumptions. Bolded numbers are associated with the modeling decisions used in the result and discussion sections.

| | Early snowmelt timing metrics | $\beta_1$: AT | $\beta_2$: Pp | $\beta_3$: RH | $\beta_4$: SWR | $\beta_5$: ATxPp | $\beta_6$: ATxRH | $\beta_7$: ATxSWR | $\beta_8$: PpxRH | $\beta_9$: PpxSWR | $\beta_{10}$: RHxSWR |
|---|---|---|---|---|---|---|---|---|---|---|---|
| r > 0.5 | 1st snowmelt day* | n/a | n/a | n/a | n/a | n/a | n/a | n/a | n/a | n/a | n/a |
| | 1st 3 consecutive snowmelt days | -0.41 | 0.74 | 0.002 | 0.38 | 0.19 | n/a | n/a | -0.33 | n/a | -0.19 |
| | DOS$_5$* | n/a | n/a | n/a | n/a | n/a | n/a | n/a | n/a | n/a | n/a |
| | DOS$_{10}$ | -0.55 | 0.45 | 0.22 | 0.56 | 0.26 | n/a | n/a | n/a | 0.23 | -0.21 |
| | DOS$_{20}$ | -0.39 | 0.46 | 0.33 | 0.68 | 0.10 | n/a | n/a | -0.10 | 0.12 | -0.28 |
| | DOS$_{30}$ | -0.32 | 0.39 | 0.38 | 0.76 | n/a | 0.06 | n/a | n/a | 0.15 | -0.27 |
| r > 0.6 | 1st snowmelt day* | n/a | n/a | n/a | n/a | n/a | n/a | n/a | n/a | n/a | n/a |
| | 1st 3 consecutive snowmelt days | -0.39 | 0.69 | 0.03 | 0.43 | 0.15 | n/a | n/a | -0.26 | 0.08 | -0.21 |
| | DOS$_5$* | n/a | n/a | n/a | n/a | n/a | n/a | n/a | n/a | n/a | n/a |
| | DOS$_{10}$ | 0.54 | 0.42 | 0.18 | 0.52 | 0.23 | n/a | n/a | n/a | 0.22 | -0.16 |
| | DOS$_{20}$ | -0.35 | 0.41 | 0.31 | 0.69 | 0.10 | n/a | n/a | -0.08 | 0.10 | -0.24 |
| | DOS$_{30}$ | -0.30 | 0.33 | 0.37 | 0.75 | 0.07 | n/a | n/a | n/a | 0.15 | -0.24 |
| r > 0.7 | 1st snowmelt day* | n/a | n/a | n/a | n/a | n/a | n/a | n/a | n/a | n/a | n/a |
| | 1st 3 consecutive snowmelt days | -0.45 | 0.69 | 0.03 | 0.46 | n/a | 0.11 | n/a | -0.16 | 0.09 | -0.23 |
| | DOS$_5$* | n/a | n/a | n/a | n/a | n/a | n/a | n/a | n/a | n/a | n/a |
| | DOS$_{10}$ | -0.46 | 0.39 | 0.20 | 0.55 | 0.21 | -0.08 | n/a | -0.09 | 0.11 | -0.17 |
| | DOS$_{20}$ | -0.31 | 0.30 | 0.36 | 0.77 | 0.10 | n/a | n/a | n/a | 0.14 | -0.24 |
| | DOS$_{30}$ | -0.29 | 0.29 | 0.38 | 0.77 | 0.08 | n/a | n/a | n/a | 0.17 | -0.26 |
| r > 0.8 | 1st snowmelt day | -0.57 | 0.41 | 0.08 | 0.34 | 0.28 | n/a | n/a | n/a | 0.21 | -0.06 |
| | 1st 3 consecutive snowmelt days | -0.35 | 0.43 | 0.26 | 0.67 | n/a | 0.09 | n/a | n/a | 0.22 | -0.27 |
| | DOS$_5$ | -0.43 | 0.39 | 0.21 | 0.56 | 0.23 | n/a | n/a | -0.09 | 0.14 | -0.19 |
| | DOS$_{10}$ | -0.34 | 0.37 | 0.28 | 0.68 | 0.16 | n/a | n/a | -0.09 | 0.13 | -0.26 |
| | DOS$_{20}$ | **-0.31** | **0.29** | **0.37** | **0.75** | **0.11** | **n/a** | **n/a** | **n/a** | **0.18** | **-0.29** |
| | DOS$_{30}$ | -0.29 | 0.29 | 0.37 | 0.76 | 0.09 | n/a | n/a | n/a | 0.18 | -0.26 |

**Table A5: Coefficient of determination ($R^2$) and slope (in parenthesis, days $°C^{-1}$) of the linear regression between the empirical diel streamflow-based model sensitivity to warming and sites' mean winter air temperature as presented in Figure 8B, for different early snowmelt day metrics and correlation cutoffs (r) between hourly solar radiation and streamflow. DOSxx represent the date when the xx$^{th}$ percentile of snowmelt days occurs. Bolded numbers are associated with the modeling decisions used in the result and discussion sections.**

| Early snowmelt timing metrics | r > 0.5 | r > 0.6 | r > 0.7 | r > 0.8 |
|---|---|---|---|---|
| First snowmelt day | 0.08 (0.61) | 0.09 (0.47) | 0.03 (0.47) | 0.23 (-0.75) |
| First 3 consecutive snowmelt days | 0.02 (-0.30) | 0.08 (-0.51) | 0.00 (-0.05) | 0.00 (-0.07) |
| DOS$_5$ | 0.00 (0.04) | 0.01 (-0.18) | 0.02 (-0.32) | 0.25 (-1.00) |
| DOS$_{10}$ | 0.00 (-0.09) | 0.25 (-0.86) | 0.37 (-1.17) | 0.2 (-0.66) |
| DOS$_{20}$ | 0.27 (-0.68) | 0.35 (-0.89) | 0.37 (-0.99) | **0.33 (-0.75)** |
| DOS$_{30}$ | 0.22 (-0.57) | 0.26 (-0.65) | 0.27 (-0.66) | 0.20 (-0.52) |

925