# Peer review of "Diel streamflow cycles suggest more sensitive snowmelt-driven streamflow to climate change than land surface modeling does"

_Hydrology and Earth System Sciences, 2021_

## Author Response (AR1)

Jessica Lundquist (Referee)

Referee comment on "Diel streamflow cycles suggest more sensitive snowmelt-driven streamflow to climate change than land surface modeling" by Sebastian A. Krogh et al., Hydrol. Earth Syst. Sci. Discuss., https://doi.org/10.5194/hess-2021-437-RC1, 2021

**Answers provided in RED**

Krogh et al. present an interesting analysis comparing climate change sensitivity impacts on streamflow in the western United States between space for time substitution (which they term STS) and more traditional modeling techniques, where they focus on NoahMP- WRF pseudo-global-warming simulations (termed PGW). They introduce a new metric based on diurnal fluctuations in streamflow that are lag-correlated with solar radiation, and then calculate the day of year when 20% of all days with well-correlated diurnal fluctuations have passed. **I like the idea and the premise of the paper, but I feel that major revisions are necessary to disentangle all the possible ways that errors in the analysis could lead to misconceptions in the results**. I also feel that the number of acronyms and metrics in the paper (STS, PGW, DOS_20, etc.) make the written text hard to follow, and **I strongly recommend that the authors minimize their use of acronyms**, perhaps provide a table of acronyms and metrics, and overall work to increase clarity. I have organized my comments into requests for Major and Minor revisions below. The authors are welcome to contact me directly if they have questions: Jessica Lundquist, jdlund@uw.edu

**Dear Professor Jessica Lundquist, we greatly appreciate your critical feedback, and we think that we have addressed your concerns in this reviewed version. Regarding the use of acronyms, which was a concerned also raised by the 2nd reviewer, we have introduced a new table with a list of acronyms (as suggested), and we are no longer using STS and PGW. However, we believe that the other acronyms are necessary to shorten the length of sentences, figure captions, figures labels etc.**

Major:

**1) You need a clear analysis of how well your diurnal-cycle-correlation metric works across a range of streams.**

1a. line 199-200 "more variable mean annual autocorrelation that ranges between roughly 0.1 and 0.6, with a mean value around 0.4" —**need to explain what different mean annual autocorrelations refer to**. These numbers are really new to most people. It would be helpful to tie this metric to the examples in Figure 1, as well as a discussion of rain vs. snow — a lot of the "snowmelt days" marked with purple circles in Figure 1 look like rain storms to me. The South Fork of the Tolt mostly gets rain, but also rain on snow. How do diurnal cycles that are identified but aren't really snow melt impact your results?

**We have removed the auto-correlation metric from the analysis as it seemed to be confusing and the associated correlations were not very strong. Regarding Figure 1, and as also noted by the 2nd reviewer, we have simplified it to improve clarity and removed the example of the South Fork of the Tolt River. Regarding the effect of rainfall on our results (a recurrent comment), we have added a new analysis that quantifies how many of the snowmelt days (from the diel analysis) occurred in days with rainfall (see new Figure 2 and lines 175-190). To assess whether it rained, we used the CAMELS precipitation database based on DAYMET and a**

**precipitation threshold of 5 mm (also tested 1 mm and 10 mm) and an air temperature threshold of 2 C. With this analysis we found that, on average, about 7% of the days classified as snowmelt also had rainfall, but this percentage was larger in the Pacific Northwest. We argue that this value is relatively small, supporting our analysis. This analysis is not perfect as it might include or exclude days with rainfall due to the nature of the datasets and the (somewhat) arbitrary selection of rainfall thresholds, but nonetheless, we believe that it provides a reasonable estimate for rainfall.**

**Additionally, we have set up several checks in our method to limit the detection of snowmelt days that could be triggered by rainfall.  First, we apply a more restricted monthly and site-specific window of lagged correlations based on clear-sky snowmelt-driven diel cycles only (section 2.2). This limits rainfall coming at a time different from typical snowmelt (or ET) causing a false positive melt day. Second, the rainstorm needs to have a specific diel cycle that will strongly correlate with solar radiation. On a complete cloudy day, solar radiation will have a diurnal cycle like a clear sky day, so a rainstorm that produces a snowmelt-like response (depending on watershed's surface and subsurface connectivity and rainfall histogram) may potentially produce a false positive. On a partly cloudy day, where rainfall occurred but either before or after the event there were clear sky conditions, the chance to have a highly correlated rainfall-induced diel cycle that is highly correlated with solar radiations is likely minimal as the shape of the solar radiation diel cycle can have several discrete changes. For these reasons, snowmelt produced by rain-on-snow should mostly be excluded from our analysis.**

1b. As an alternate approach to when snowmelt is significant, you could look at the power spectra of your time series. See Figure 6 in Lundquist and Cayan 2002. The days with a sharp increase in power at the once per day cycle indicate snowmelt, whereas rain exhibits a much more red spectra. I know that power spectra are commonly used by oceanographers and not hydrologists, so your method is likely easier to understand, but it would be nice to have an independent method to check.

**We appreciate the recommendation of the reviewer about the power spectra, but as detailed in the previous answer, we have taken a different direction to show whether rainfall occurred or not during our classified snowmelt days. We think this analysis will be more intuitive to hydrologists or snow scientists.**

1c. In particular, I recommend clearer discussion about the strengths and weaknesses of this approach. It will miss rain-on-snow (signal dominated by rain), as well as early melt into dry soil (no streamflow response). It may also misclassify rain with a diurnal structure to it as snowmelt. Therefore (and you allude to this multiple times in the manuscript but should make it clearer), the method is best at detecting melt in non-rainy locations with fairly-saturated soils. With that in mind, which of your basins do you trust the signal the most.

**The reviewer makes good points about what the method can and cannot do. We have improved our discussion as suggested by the reviewer and added more details about the strengths and weaknesses of our approach (see lines 304-305, and 350-355), where we discuss potential problems with rain-on-snow and our new analysis that checks for the effect of rainfall. We appreciate the reviewer's comments as they have more clearly shown why the two outlier catchments behave differently (they receive more rain as expected).**

1d. Section 3.1 explains how well the DOS_20 is related to simpler magnitude metrics (DOQ_25 and DOQ_50) but doesn't really justify why the DOS_20 is helpful beyond those metrics — can you better explain what we gain by doing this extra analysis. This section also identifies some rain-dominated rivers wherein these metrics appear less correlated. Is this because the method breaks down? Or can we learn important information from this change in relationship?

**$DOS_{20}$ aims to capture snowmelt-streamflow connectivity; however, it does not imply anything about the contribution (volume) of snowmelt to streamflow. As such, we propose that this metric can be implemented as a relatively easy way to benchmark hourly hydrological and land-surface models beyond typical daily streamflow metrics or point-scale continuous SWE measurements. Specifically, we see potential to use this information to validate snowmelt dynamics of a model.**

**About the value of section 3.1, we believe there are two key points to be stated. First, the diel method is more uncertain under rainier conditions as it may potentially misclassify snowmelt events due to rainfall (see new Figure 2 for a quantification of this effect), and second, under rainier conditions the timing of streamflow volume is likely to be more strongly controlled by the timing of rainfall as opposed to the timing of snowmelt, and thus those sites deviate from the 1:1 line in DOS20 vs DOQ25 and DOQ50.**

**2) You need to more explicitly discuss the difference between a stream's climate sensitivity of snowfall changing to rainfall vs. a climate sensitivity of earlier snowmelt.**

2a. Many of the earlier papers on streamflow sensitivity to climate change highlighted basins in the transitional rain-snow zone as being most sensitive because snowfall shifts to rainfall. From my own experience, the diurnal cycle in streamflow is particularly hard to detect in these basins because rain-induced runoff is such a larger signal than snow-induced runoff, especially when both happen more or less at the same time. Therefore, I imagine that your snowmelt index uniquely does not work well in these basins (e.g., the Tolt example in your paper, or the NF American River example in Lundquist and Cayan 2002 Fig. 6). I could imagine that for these basins, you could even get DOS_20 moving later in the season with warming if early season events are all rain and only a later, non- rainy period exhibits snowmelt.

**We agree that a better discussion of the effects of changes from snow to rain on our results is merited, and we have improved it to include these points (e.g., lines 495-505). As mentioned above, we include several filters to minimize the selection of rainfall-driven diel cycles (quantified by the new analysis, Figure 2). Although we don't find any site with a later DOS20, as potentially identified by the reviewer, we do find that more rainfall-dominated watersheds have a smaller sensitivity in DOS20 to climate change (Figure 8).**

2b. I imagine that including rain-on-snow or rain-dominated basins would bias your correlations with humidity because these tend to be more humid basins but also may have spurious results.

**We tried to maximize the site and inter-annual variability in the dataset to increase the predictive power of the space-for-time approach, as historically cold sites will transition into warmer and more humid sites and into those with rainier conditions. That being said, we recognize the challenges in reliably capturing snowmelt events where rainfall is important (as discussed in previous major comment) and correctly identified by the reviewer. It's relevant to highlight that those sites in the Pacific Northwest (#24, 25 and 31) that have low snowfall contributions (as highlighted in Figure 5) are ultimately not used for the streamflow sensitivity analysis, and thus do not impact the conclusions. Nonetheless, we recognize the good point raised by the reviewer and we clarified and discussed it in the revised version of the manuscript.**

**2c. I encourage the authors to think about rainfall vs snowfall and snowmelt sensitivities separately and to decide if they want to address both in this paper or only focus on the latter. Then, be very clear about this decision in the paper discussion.**

**It is not easy to disentangle the two, but we agree that our method is better suited to answer questions about snowmelt sensitivity and that should be the focus of the paper. However, we recognize our empirical analysis reflects both the effect of changing precipitation partitioning and snowmelt sensitivities. We improved the discussion to address this comment (see lines 380-390).**

**3) You need to more clearly evaluate how well your NoahMP-WRF model set up is simulating streamflow timing in the current climate before examining the results of its climate sensitivity.**

3a. It appears that you have a biased simulation of NoahMP-WRF — if the historic runoff date is off by 50 days (see line 260), the model is either simulating too much rain and too little snow or melting snow way too early. It's hard to draw conclusions on sensitivity when using a biased model. Of course, if the model has less snow than the real world, it will be less sensitive to that snow disappearing. The paper would be much more meaningful if you included some evaluation of your NoahMP-WRF simulations — how do they compare to baseline observations and to other models run over the domain (similar western US climate-change papers).

**The reviewer makes a good point and we have improved and highlighted better the description of the model performance (lines 410-420). We do include an evaluation of DOQ25 and DOQ50 in the paper (Figure 9), where we show that DOQ25 is more biased than DOQ50; however, we do acknowledge that it is not a comprehensive comparison. Detailing the exact biases of NoahMP simulations in the past is beyond the scope of this study, but we have detailed previous efforts in this arena. Just to clarify, these simulations made by the National Center for Atmospheric Research (NCAR) presented by (Liu et al., 2017) have been previously tested in terms of their meteorology and snow components (Liu et al., 2017; Scaff et al., 2020). We do agree with Dr. Lundquist in that one should make sure the model reliably represents a particular system before looking at its sensitivity to climate change. Nonetheless, these types of simulations have been used for climate change analyses (Musselman et al., 2017, 2018), but their runoff components have not been tested to our knowledge. Furthermore, the NoahMP model underlies the US National Water Model**

([https://water.noaa.gov/about/nwm](https://water.noaa.gov/about/nwm)) **and thus is highly relevant to both policy and research.**

3b. Also, if the NoahMP-WF simulations perform better in certain regions (if I'm correct, these were only carefully vetted for Colorado), you may also want to focus your analysis on those regions separately. Do you get closer agreement in areas where the model represents snow processes more accurately? Might a check for space-for-time sensitivity against model sensitivity be a good check for model fidelity?

**For the historical DOQ25 the NoahMP-WRF model actually performed the best in rainier sites (see circled blue symbols in Figure 9a) and a few other sites classified as 'cloudy' and 'partly cloudy', whereas the Rocky Mountain sites, characterized by 'sunny' snowmelt events, were among the most biased (see blue filled circles in Fig9a). This suggests that the timing of streamflow volume is better represented in areas where snowmelt processes are less important, though other variables like topographic (and thus climatic) gradient can also be important.**

4. Discussion should be better streamlined and organized. This may be a good place to address major comments 1-3 above.

**We have improved the discussion based on Dr. Lundquist suggestions, which will hopefully address her main concerns.**

Minor:

Abstract: 1st sentence, "may cause" — I think the literature is pretty conclusive that warming does cause snow to melt earlier. Abstract should define what you mean by the 20th percentile of snowmelt days — this is meaningless to someone only reading the abstract. What do you mean by colder places are more sensitive than warmer places? In what way? Earlier snowmelt? If there's no snow, of course it wouldn't be sensitive to that.

**We have changed the abstract to read "climate change will cause ...". We have changed the abstract to provide a better description of DOS20 and what we mean by the different sensitivities.**

Line 120: "DAYMET dataset (daymet.ornl.gov), which in turn is based on ground observations" — it's interpolated from existing ground observations — worth specifying as sometimes this is far from truth.

**We have changed to read as suggested by the reviewer.**

lines 202-205 The percent of streamflow volume by a certain date vs temperature has been well established in the early literature (Stewart et al. 2005). Also see Lundquist et al. 2004 for a review of different ways to define the "spring onset" from snow pillows and from a hydrograph: https://doi.org/10.1175/1525-7541(2004)005<0327:SOITSN>2.0.CO;2

**We appreciate the references. We have included them in lines 417-422.**

line 215: Yes, these sites are low elevation, receiving primarily rain, and I think your methodology is identifying rain events as having a diurnal cycle.

**As previously mentioned, our method cannot guarantee that rainfall-induced cycles are not picked up (as hourly rainfall data would be required); however, we have implemented several filters to rule out such cases. The new analysis (Figure 2) shows that those watersheds have 22%, 15% and 29% of snowmelt days with rainfall, suggesting that in some cases our method is including rainfall-driven diel cycles. However, it is important to highlight that those watersheds are ultimately not included in the streamflow sensitivity analysis and the comparison against NoahMP-WRF. An improved discussion for the effect of rainfall on our method is now included in the discussion section.**

line 259: "greatly underestimated" — I think you mean than it's modeled as earlier than observed, right? Underestimated makes me think that the magnitude of the streamflow is too low.

**We mean that the date DOQ50 is underestimated by the model, but to avoid confusions we will change it to "earlier than observed" as suggested by Dr. Lundquist.**

**References:**

**Liu, C., Ikeda, K., Rasmussen, R., Barlage, M., Newman, A. J. A. J. A. J. A. J., Prein, A. F. A. F., Chen, F., Chen, L., Clark, M., Dai, A., Dudhia, J., Eidhammer, T., Gochis, D., Gutmann, E., Kurkute, S., Li, Y., Thompson, G. and Yates, D.: Continental-scale convection-permitting modeling of the current and future climate of North America, Clim. Dyn., 49(1–2), 71–95, doi:10.1007/s00382-016-3327-9, 2017.**

**Musselman, K. N., Clark, M. P., Liu, C., Ikeda, K. and Rasmussen, R.: Slower snowmelt in a warmer world, Nat. Clim. Chang., 7(3), 214–219, doi:10.1038/nclimate3225, 2017.**

**Musselman, K. N., Lehner, F., Ikeda, K., Clark, M. P., Prein, A. F., Liu, C., Barlage, M. and Rasmussen, R.: Projected increases and shifts in rain-on-snow flood risk over western North America, Nat. Clim. Chang., 8(September), doi:10.1038/s41558-018-0236-4, 2018.**

**Scaff, L., Prein, A. F., Li, Y., Liu, C., Rasmussen, R. and Ikeda, K.: Simulating the convective precipitation diurnal cycle in North America's current and future climate, Clim. Dyn., 55(1–2), 369–382, doi:10.1007/s00382-019-04754-9, 2020.**

RC2

**Answers provided in red**

The authors present a new means of considering the sensitivity of snowmelt timing and streamflow response under warming climate conditions based on space for time substitutions. Their metric (DOS_20) is based on diel fluctuations in streamflow that correlate with solar radiation (after a time lag of 6-18 hours). They use this metric to assess regional sensitivity to warming across an array of small montane basins in the western U.S. They compare their approach to one using a physically-based modeling framework, highlighting differences in snowmelt-streamflow sensitivities derived from each method.

I think the approach presented here can provide valuable insights into the implications climate warming holds for water forecasting and management. However, I found the paper somewhat difficult to follow. I believe significant revisions are necessary to improve the clarity of the analysis. These are enumerated below.

**We greatly appreciate the positive feedback.**

1. Devote more space to background information. Numerous concepts are discussed with minimal introduction (e.g. space for time substitution, mean annual autocorrelation, diel streamflow cycles, etc). I understand that the authors are snow hydrologists writing for other snow hydrologists, but the paper would be significantly easier to follow with a proper setup for many of the concepts being discussed.

**We appreciate the reviewer´s feedback. We devote an entire paragraph in the introduction that explains the space-for-time approach and differences with more traditional hydrological modeling tools. We have removed the autocorrelation metrics as they seemed to be confusing as also noted by the 1st reviewer. We have extended the description of diel cycles in the introduction.**

2. Streamline extremely dense figures and captions. There is a ton of information included in each figure--particularly Figures 1-3. I think it would be beneficial to break some of these into multiple figures in order to make them more digestible. At the very least, the authors should consider changes such as increasing the font size (overall, but particularly in the tiny inset histograms) and increasing the clarity of the captions, even if that means making them longer. It took me a long time to understand that the "thick line" referenced in the Figure 1 caption referred to the border of the text box itself.

**We agree with the reviewer, and we have substantially changed the figures to make them easier to follow and understand. For example, we removed unnecessary text in Figure 1 (and the "thick line" classification) and removed two panels. Previous Figures 2 and 3 were divided into two figures each, and are now new Figures 3, 4, 5 and 6 (as we added a new Figure 2). Overall, we increased font size, and removed the inset histograms from plots (which became new panels).**

3. Reduce the number of abbreviations in the text. Overall, there are a lot of abbreviations in this manuscript. Certain sections (e.g. Section 3.3) are particularly dense with abbreviations, and correspondingly hard to follow. I would recommend cutting down on the number of abbreviations for clarity.

**This comment was also provided by the 1st reviewer. We removed two acronyms (STS and PGW) and provide a new table with a list of abbreviation as suggested by reviewer 1. However, we are keeping some key abbreviations (DOS20, DOQ25 and DOQ50) to keep the overall text shorter.**

4. Elaborate on the NoahMP-WRF simulations. It's hard to draw conclusions on this section of the analysis, because relatively little information is given about these simulations. An important feature of NoahMP is that it has multiple options for simulating rain-snow partitioning and snowpack albedo. It also has multiple snowpack-related parameters to which both snow and streamflow are quite sensitive. Without knowing the model physics options and parameters used, it is difficult to conclude whether the biases the authors observed is a structural problem with the model or just a poor setup.

We have added more details about the improvement made by Liu et al (2017) to Noah-MP to better represent snowpack processes (see line 197-203). Simulations by Liu et al., have been tested for precipitation and snowpack dynamics by Scaff et al. (2020) and Liu et al (2017) and used to investigate the impact of climate change on snowpack (Musselman et al., 2017, https://www.nature.com/articles/nclimate3225 ). We have also included more details about the performance of model runs and other studies using the same runs for better context in the discussions (lines 410-420).

5. Rain on snow. This seems like an important point to discuss in a paper about snowpack and streamflow under climate warming. How well does this new metric handle rain-on-snow events? Can they be resolved and included/excluded? Or are they a confounding factor?

As also noted by reviewer 1, rain-on-snow events are problematic in our method as we have no explicit way to address the impact of rainfall due to lack of reliable hourly rain/snow observations. As our method implements several filters to avoid the impact of rainfall-induced streamflow cycles, it will also (very likely) miss rain on snow events. We have improved the discussion to incorporate the difficulties of this method to incorporate rain on snow events (line 305, and 350-355). Additionally, we are including a new analysis (new figure 2) to quantify whether rainfall occurred during snowmelt days, based on daily precipitation (from CAMELS and DAYMET) and an air temperature threshold. This analysis shows that, on average, only 7% of the days classified as snowmelt also had rain > 5 mm. This was also added to the discussion (line 350-355) to provide better context for the challenges associated with this method.

**References**

Liu, C., Ikeda, K., Rasmussen, R., Barlage, M., Newman, A. J. A. J. A. J. A. J., Prein, A. F. A. F., Chen, F., Chen, L., Clark, M., Dai, A., Dudhia, J., Eidhammer, T., Gochis, D., Gutmann, E., Kurkute, S., Li, Y., Thompson, G. and Yates, D.: Continental-scale convection-permitting modeling of the current and future climate of North America, Clim. Dyn., 49(1–2), 71–95, doi:10.1007/s00382-016-3327-9, 2017.

Scaff, L., Prein, A. F., Li, Y., Liu, C., Rasmussen, R. and Ikeda, K.: Simulating the convective precipitation diurnal cycle in North America's current and future climate, Clim. Dyn., 55(1–2), 369–382, doi:10.1007/s00382-019-04754-9, 2020.

---

## Author Response (AR2)

Sebastian,

I appreciate your detailed and thoughtful response to my earlier review, and I think that the issue of rain signals has now been much better addressed. However, as I read the paper again, I find the term "space-for-time" to be very confusing throughout, and I believe the paper will have a much better impact on the community if it is easier to read and follow. I recommend that you drop the terminology "space-for-time" and instead explain clearly what you are doing in the regressions. As I understand it, you are fitting regressions to watershed temperature and precipitation and other variables across multiple sites and then using data from warmer sites to predict future changes in currently colder sites. If this is not what you're doing, then my personal confusion is just an example of how the paper might be confusing, and please clarify what you are doing. If that's part of it, and there's more to it, please also explain that. This is particularly critical in the abstract and the introduction because I could not figure out what the main goal of the paper is after reading the abstract and Section 1. When I'm reading papers in general (not as an assigned reviewer), I'm very likely to stop reading if I can't figure out the objectives by that point. If you want to use the term "space-for-time", please make sure that it is clearly defined both in the abstract (if used there) and in the first section of the paper. I would be happy to read this again after you have rewritten for clarity. I think there's a lot of interesting analysis here, but without making the paper clearer, that analysis will be not be useful to the scientific community.

Thank you, Jessica Lundquist

Dear Professor Jessica Lundquist,
We appreciate your previous comment about the issues related to the rain signal and we are happy that our response was satisfactory.
Your interpretation of our method is correct, but given the lack of clarity following your comment, we have decided to drop the term space-for-time (which we thought was better established in the community than it is). Instead, we now call it an "empirical diel streamflow-based model" everywhere in the text. As you correctly interpreted, this model is based on the stepwise multiple linear regression that uses 4 climate variables (precipitation, temperature, humidity, and solar radiation; predictors) to predict the DOS20 metric. We then use those DOS20 predictions and the linear regression between DOS20 and DOQ25 and DO50 to project changes in streamflow volume timing. Ultimately, this is an empirical model, which explains our name of choice. We emphasize and clarify this in section 2.3 now called "The empirical diel-streamflow based model".

Regarding the main goal of the paper, we agree that it wasn't clear and we have changed both the abstract and the introduction to be as clear as possible, for which we have added the following sentences to the abstract (lines 15-20):

"Quantifying how sensitive is streamflow timing to climate change, and in which places it is the most sensitive, remains a key question. Physically based hydrological models are often used for this purpose; however, they have embedded assumptions that translate into uncertain hydrological projections. Such uncertainties need to be quantified and constrained (as possible) to provide reliable projections. The purpose of this study is to evaluate differences in projected changes to streamflow volume timing by the end of the century between a new empirical model based on diel (daily) streamflow cycles and regional land-surface simulations across the mountainous western US."

Additionally, we have added the following to the Introduction section (lines 106-110):

"However, it remains unknown whether information embedded in the diel streamflow response following snowmelt events can be used to inform streamflow predictions due to climate change, and whether such projections are consistent with current land-surface simulations. The purpose of this research is to evaluate potential differences in projected changes to streamflow volume timing by the end of the century between a new empirical diel streamflow-based model and regional land-surface simulations across mountainous western US headwater catchments"

We believe that clarifies the main purpose of the study.

Minor details:
Abstract should be stand alone and understandable by someone who has not read the whole paper. This is not true right now. Please, for greater impact, rewrite the abstract.

We have changed the abstract significantly to make both the main goal and methods clearer.

line 23: space-for-time substitution used in abstract but not defined — please clarify what you mean by this. I recommend not even using that term but instead just write out what specifically you are trying to do.

As suggested, we are dropping the term space-for-time and now using "empirical diel streamflow-based model".

Line 66 on: explanation of space for time is still confusing. Here, what is space and what is time. I think you mean that you assume that warmer locations today represent currently cooler locations tomorrow, is that right? Please clarify.

That is correct, and we have changed the sentence to read as followed (lines 73-80) to avoid further confusions:

"Empirical models assume that long-term and often site-to-site statistical relationships among predicting variables (e.g., precipitation and air temperature) and water fluxes (e.g., evapotranspiration and streamflow) can be used to understand and model their likely changes over time or space. Empirical models used to predict changes over time (sometimes referred to space-for-time substitutions) have been used in fields such as hydrology (Goulden and Bales, 2014; Jepsen et al., 2018; Sivapalan et al., 2011), biodiversity (Blois et al., 2013) and tree growth (Klesse et al., 2020) to predict responses to climate change. Such models use information from different places ("space"), typically spanning a wide range of conditions (e.g., climate gradient), to predict changes over time. For example, observed characteristics from warm regions maybe used to infer future changes in cold regions due to global warming."

line 93: "driven by either snow or ice melt and evapotranspiration " I think you need to switch the "or" and the "and" in that sentence.

We have changed the "or" to "and" as suggested by the reviewer.

Line 188: still very confusing use of space for time. Just say what part of the regression equation you're changing. The temperature and precip inputs? Something else?

We have dropped the space-for-time terminology and clarify what we mean about or DOS20 predictions. Please see changes in lines 194-195, where we specify that:

"The MLR model is the basis of our empirical diel streamflow-based model, which is used to assess changes in DOS20 due to climate change (i.e., changes in x1, x2, x3 and x4 in Eq. (1))."

References:

Blois, J. L., Williams, J. W., Fitzpatrick, M. C., Jackson, S. T. and Ferrier, S.: Space can substitute for time in predicting climate-change effects on biodiversity, Proc. Natl. Acad. Sci., 110(23), 9374–9379, doi:10.1073/pnas.1220228110, 2013.
Goulden, M. L. and Bales, R. C.: Mountain runoff vulnerability to increased evapotranspiration with vegetation expansion, Proc. Natl. Acad. Sci., 111(39), 14071–14075, doi:10.1073/pnas.1319316111, 2014.
Jepsen, S. M., Harmon, T. C., Ficklin, D. L., Molotch, N. P. and Guan, B.: Evapotranspiration sensitivity to air temperature across a snow-influenced watershed: Space-for-time substitution versus integrated watershed modeling, J. Hydrol., 556, 645–659, doi:10.1016/j.jhydrol.2017.11.042, 2018.

Klesse, S., DeRose, R. J., Babst, F., Black, B. A., Anderegg, L. D. L., Axelson, J., Ettinger, A., Griesbauer, H., Guiterman, C. H., Harley, G., Harvey, J. E., Lo, Y., Lynch, A. M., O'Connor, C., Restaino, C., Sauchyn, D., Shaw, J. D., Smith, D. J., Wood, L., Villanueva-Díaz, J. and Evans, M. E. K.: Continental-scale tree-ring-based projection of Douglas-fir growth: Testing the limits of space-for-time substitution, Glob. Chang. Biol., 26(9), 5146–5163, doi:10.1111/gcb.15170, 2020.

Sivapalan, M., Yaeger, M. A., Harman, C. J., Xu, X. and Troch, P. A.: Functional model of water balance variability at the catchment scale: 1. Evidence of hydrologic similarity and space-time symmetry, Water Resour. Res., 47(2), 1–18, doi:10.1029/2010WR009568, 2011.

---

## Author Response (AR3)

Dear Editor,

We appreciate the opportunity to improve our manuscript following your minor revision. We are including the reviewed version, where we have made significant changes to make it easier to follow and read. In the attached document you will find a re-organized discussion section and many grammar changes, all of which were revised by the authoring team, including senior scientists Dr. James Kirchner and Dr. Adrian Harpold.
Best regards

Sebastián Krogh